# Transcription elongation factor AFF2/FMR2 regulates expression of expanded GGGGCC repeat-containing *C9ORF72* allele in ALS/FTD

Yeliz Yuva-Aydemir[1,3], Sandra Almeida [1,3]*, Gopinath Krishnan[1], Tania F. Gendron[2] & Fen-Biao Gao [1]*

Expanded GGGGCC ($G_4C_2$) repeats in *C9ORF72* cause amyotrophic lateral sclerosis (ALS) and frontotemporal dementia (FTD). How RNAs containing expanded $G_4C_2$ repeats are transcribed in human neurons is largely unknown. Here we describe a *Drosophila* model in which poly(GR) expression in adult neurons causes axonal and locomotor defects and premature death without apparent TDP-43 pathology. In an unbiased genetic screen, partial loss of Lilliputian (Lilli) activity strongly suppresses poly(GR) toxicity by specifically down-regulating the transcription of GC-rich sequences in *Drosophila*. Knockout of *AFF2/FMR2* (one of four mammalian homologues of Lilli) with CRISPR-Cas9 decreases the expression of the mutant *C9ORF72* allele containing expanded $G_4C_2$ repeats and the levels of repeat RNA foci and dipeptide repeat proteins in cortical neurons derived from induced pluripotent stem cells of *C9ORF72* patients, resulting in rescue of axonal degeneration and TDP-43 pathology. Thus, AFF2/FMR2 regulates the transcription and toxicity of expanded $G_4C_2$ repeats in human *C9ORF72*-ALS/FTD neurons.

[1] Department of Neurology, University of Massachusetts Medical School, Worcester, MA 01605, USA. [2] Department of Neuroscience, Mayo Clinic Florida, Jacksonville, FL 32224, USA. [3] These authors contributed equally: Yeliz Yuva-Aydemir, Sandra Almeida. *email: Sandra.almeida@umassmed.edu; fen-biao.gao@umassmed.edu

Amyotrophic lateral sclerosis (ALS) and frontotemporal dementia (FTD) are pathologically and genetically linked neurodegenerative diseases that share common dysregulated molecular pathways[1,2]. ALS causes progressive loss of motor neurons and muscle weakness[3], whereas FTD causes focal atrophy in the frontal and temporal lobes, resulting in progressive cognitive and behavioral alterations and other deficits[4]. In most cases of ALS and FTD, the DNA/RNA-binding protein TDP-43 (transactive response (TAR)-DNA-binding protein 43) is the major constituent of pathological aggregates in patient neurons[5]. Moreover, the GGGGCC ($G_4C_2$) repeat expansion in the first intron of *C9ORF72* is the most common genetic cause of both ALS and FTD[6,7], further highlighting the shared pathogenic mechanisms in these disorders.

In *C9ORF72*-ALS/FTD, three pathogenic mechanisms of neurodegeneration have been proposed: partial loss of C9ORF72 function, toxicity from bidirectionally transcribed repeat RNAs, and dipeptide repeat (DPR) proteins induced toxicity[8]. Five DPR proteins—poly(GP), poly(GA), poly(PA), poly(GR), and poly(PR)—are found in the brains of patients with *C9ORF72*-ALS/FTD[9–11]. Of these proteins, poly(GR) expression strongly correlates with neurodegeneration in *C9ORF72*-ALS/FTD[12,13]. Although the underlying mechanisms are not fully understood, poly(GR) expression per se can cause neurotoxicity in vitro and in vivo[14–21]. In pull-down experiments, proteins that interact with arginine-containing DPR proteins include cytoplasmic and mitochondrial ribosomal proteins, as well as other RNA-binding proteins such as components of stress granules with low complexity domains, spliceosomes, and nucleoli[19,22–27]. Consistent with these findings, poly(GR) has been linked to defects in nuclear cytoplasmic transport, translation, stress granule dynamics, nucleolar function, splicing, mitochondrial function, DNA damage response, and other cellular processes[1,2].

Accumulating evidence that DPR proteins are toxic highlights the importance of understanding the mechanisms underlying the toxicity and production of DPR proteins. To explore these issues, we established a *Drosophila* poly(GR) model in which 80 repeats of GR are expressed in all adult neurons in a temporally and spatially controlled manner. We did an unbiased genetic screen in *Drosophila* and also used the CRISPR-Cas9 technology to carry out genetic manipulations in neurons derived from *C9ORF72* induced pluripotent stem cells (iPSCs). Our findings uncover a role for AFF2/FMR2, a component of the super elongation complex (SEC)-like 2, in controlling the expression and toxicity of the *C9ORF72* mutant allele containing expanded $G_4C_2$ repeats in ALS/FTD.

## Results

**Poly(GR) causes locomotor defects and death of adult flies.** Overexpression of poly(GR) driven by the *UAS-Gal4* system in the developing *Drosophila* eye is highly toxic[15–17]. To study age-dependent effects of poly(GR) expression in the adult brain, we expressed, in all postmitotic neurons of adult flies, a $(GGXGCX)_{80}$ DNA construct (X being randomly any of the four nucleotides) that contains either the ATG start codon to encode flag-tagged $(GR)_{80}$ or the TAG stop codon as the control[17] (Fig. 1a). We used the temporal and regional gene expression targeting (TARGET) system[28], in which temperature-sensitive Gal80 ($Gal80^{ts}$) inhibits Gal4-mediated transcription at low temperatures (e.g., 18 °C) but not at higher temperatures (e.g., 25 or 29 °C). *Elav-Gal4*, $Gal80^{ts}$, and $UAS$-$(GR)_{80}$ were recombined

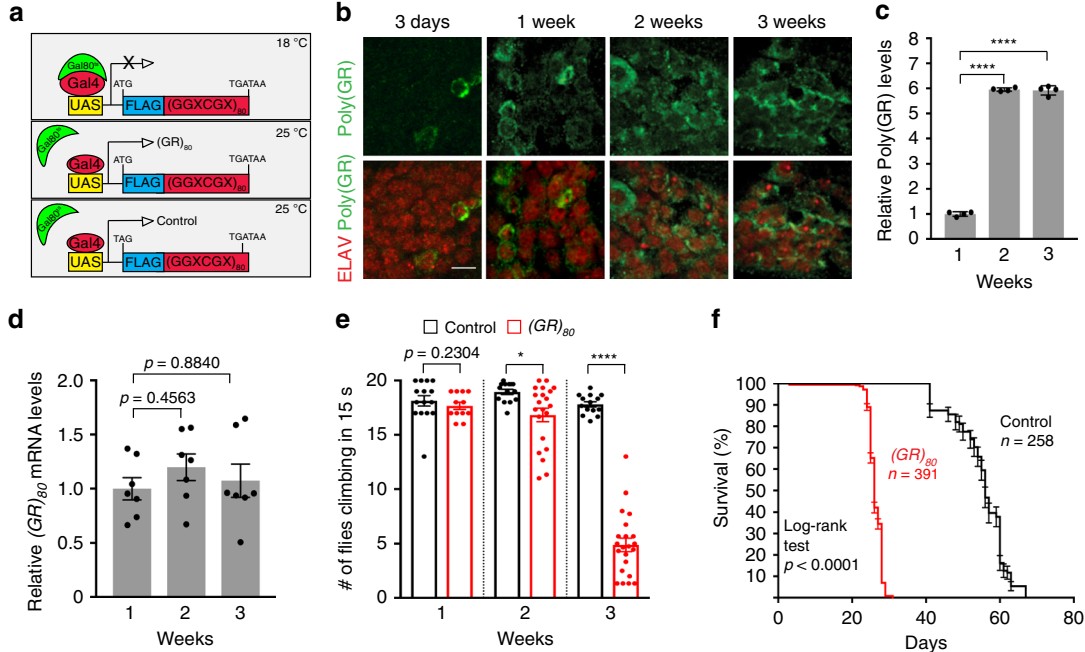

**Fig. 1** Establishment of a *Drosophila* model of *C9ORF72*-ALS/FTD. **a** Schematic of the fly model in which $Gal80^{ts}$ inhibits Gal4-mediated transcription of $(GR)_{80}$ at 18 °C, thus poly(GR) is expressed in a spatially and temporally controlled manner. $(GR)_{80}$ is encoded by $(GGXGCX)_{80}$, where X is any one of the four nucleotides. **b** Immunostaining of poly(GR) in ELAV-positive neurons of flies aged 3 days to 3 weeks old. Scale bar: 5 µm. **c** Meso Scale Discovery (MSD) ELISA analysis of poly(GR) levels in lysates of heads from 1 to 3-week-old flies. Each data point is from an independent cross, $n = 4$ independent crosses. Values are mean ± s.e.m. ****$p < 0.0001$ (one-way ANOVA, Dunnett's test). **d** qPCR analysis of the $(GR)_{80}$ mRNA level in the brain of 1–3-week-old flies with pan-neuronal expression of $(GR)_{80}$. $n = 7$ independent genetic crosses and mRNA measurements. Values are mean ± s.e.m. (one-way ANOVA, Dunnett's test). **e** Control and $(GR)_{80}$ flies at the indicated ages (one dot = 20 flies) were analyzed with the negative geotaxis climbing assay. $n = 15, 12, 14, 21, 14,$ and 22 assays for control and $(GR)_{80}$ flies at 1, 2, and 3 weeks, respectively. Values are mean ± s.e.m. *$p < 0.05$; ****$p < 0.0001$ (Mann–Whitney test). **f** Survival analysis of control and $(GR)_{80}$ flies grown at 25 °C. Source data are provided as a Source Data file.

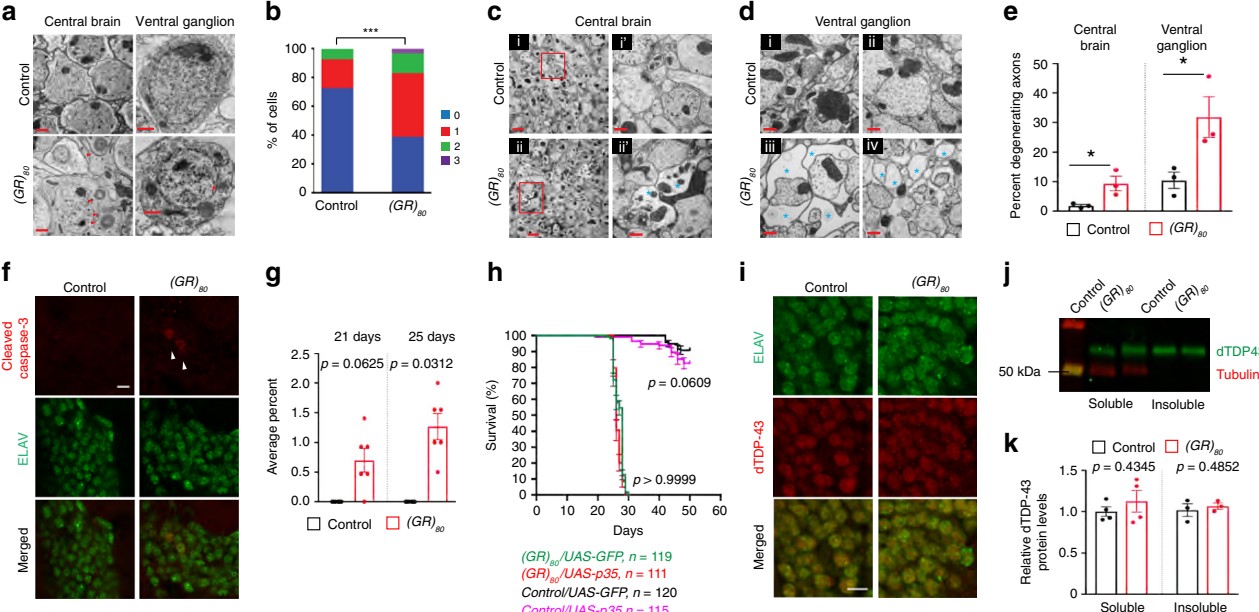

**Fig. 2** Poly(GR) causes axonal defects without TBPH pathology. **a** $(GR)_{80}$ expression leads to the accumulation of multilamellar bodies (MLBs) and axonal degeneration in *Drosophila*. Transmission electron microscopy (EM) analysis of the brain in 3-week-old control and $(GR)_{80}$ flies. Red arrowheads indicate MLBs. Scale bars: 1 μm (central brain), 0.5 μm (ventral ganglion). **b** Number of MLBs per neuron in the brain of 3-week-old flies. $n = 55$ neurons from three control flies and $n = 95$ neurons from three $(GR)_{80}$ flies were counted in a blinded manner. ***$p < 0.001$ (Chi-square analysis). **c** EM analysis of the brain of 3-week-old control and $(GR)_{80}$ flies. Red boxes in i and ii are enlarged in i' and ii'. **d** EM analysis of ventral ganglions of 3-week-old control and $(GR)_{80}$ flies. Asterisks indicate the axons without vesicles and filaments in iii and iv. Scale bars: 2 μm in i and ii, 0.5 μm in i' and ii'. **e** Quantification of axonal degeneration in the brain and ventral ganglion. Swollen axons, axons with vacuoles and damaged mitochondria and gaps in the neurophile are quantified as degenerative axon ($n = 3$ flies per genotype quantified in a blinded manner). Values are mean ± s.e.m. *$p < 0.05$ (Student's $t$ test). **f** Immunostaining for cleaved caspase-3 (red) in ELAV-positive brain neurons (green) of 21-day-old control and $(GR)_{80}$ flies. Arrowheads indicate apoptotic cells. Scale bar: 5 μm. **g** Number of neurons positive for cleaved caspase-3 in 21- and 25-day-old control and $(GR)_{80}$ central brains. $n = 6$ brains per genotype. Dots represent the percentage of average cleaved caspase-3-positive neurons out of 200–250 neurons in each fly. Values are mean ± s.e.m. *$p < 0.05$ (Wilcoxon signed-rank test). **h** Survival curve of $(GR)_{80}$ flies expressing GFP or p35. The $p$ value was determined by log-rank test. **i** Immunostaining of ELAV-positive brain neurons (green) for TBPH (red) in 21-day-old control and $(GR)_{80}$ flies. **j** Western blot analysis of soluble and insoluble proteins in the brains of 3-week-old control and $(GR)_{80}$ flies. **k** Quantification of western blot experiments ($n = 3$ independent blots). Soluble: $p = 0.4345$, Insoluble: $p = 0.4852$ (Welch's $t$ test). Source data are provided as a Source Data file.

onto the second chromosome, and *Elav-Gal4, Gal80^ts, UAS-$(GR)_{80}$/+* pupae were kept at 18 °C to prevent poly(GR) expression. Newly eclosed adult flies were kept at 25 or 29 °C to induce poly(GR) expression (Fig. 1a).

Pan-neuronal expression of $(GR)_{80}$ at 29 °C led to the death of all adult flies within 2–3 weeks (Supplementary Fig. 1a). Therefore, subsequent experiments were done at 25 °C to reduce relative $(GR)_{80}$ expression. At 25 °C, $(GR)_{80}$ expression driven by *ELAV-Gal4* became detectable in adult brain neurons at 3 days of age, and was mostly localized in the cytoplasm (Fig. 1b). This finding is consistent with the subcellular location of ectopically expressed $(GR)_{80}$ we observed in *Drosophila* motor neurons[17], human iPSC-derived motor neurons[19], and mouse brain neurons[21]. Poly(GR) expression increased gradually in an age-dependent manner, peaking at 2–3 weeks of age as shown by immunostaining (Fig. 1b) or quantification based on our recently established Meso Scale Discovery (MSD) immunoassay[21] (Fig. 1c). However, $(GR)_{80}$ mRNA levels did not change significantly (Fig. 1d), suggesting that $(GR)_{80}$ protein accumulates with aging, likely reflecting inefficient clearance. A negative geotaxis assay of control and $(GR)_{80}$ flies at 1, 2, and 3 weeks of age revealed that motor function of $(GR)_{80}$ flies began to deteriorate at 2 weeks of age and was significantly worse at 3 weeks of age (Fig. 1e). Most of the $(GR)_{80}$ flies died at 3–4 weeks of age (Fig. 1f). The compromised survival was likely due to in a large part motor neuron defects, as motor neuron-specific expression of $(GR)_{80}$ in adult flies also led to premature death (Supplementary Fig. 1b).

**Poly(GR) causes axonal defects without TDP-43 pathology.** To examine poly(GR) toxicity at the ultrastructural level, we examined 3-week-old fly brains by electron microscopy (EM). The number of neurons with multilamellar bodies (MLBs) was significantly increased in both the central brain and ventral ganglion of $(GR)_{80}$ flies (Fig. 2a, b). MLBs frequently accumulate in neurodegenerative disease, possibly reflecting defective clearance of autophagic vacuoles. Indeed, expression of FTD-associated mutant CHMP2B also results in the accumulation of MLBs that are positive for LC3II, a key protein in autophagosome formation[29]. Signs of axonal degeneration were also present, including loss of axonal integrity as shown by swollen axons with inclusions and damaged mitochondria and loss of axonal filaments in the neuropil of 3-week-old $(GR)_{80}$ flies (Fig. 2c, e and Supplementary Fig. 2).

Despite the axonal degeneration, only 0.7% of $(GR)_{80}$-expressing neurons in 21-day-old $(GR)_{80}$ brains were apoptotic and no apoptotic cells were seen in control fly brains, as shown (Fig. 2f) and quantified (Fig. 2g) by positive immunostaining for cleaved caspase-3. At 25 days of age, only 1.3% of neurons in $(GR)_{80}$ brains were positive for cleaved caspase-3 (Fig. 2g). This low level of apoptotic neuronal cell death was probably not responsible for the reduced survival of $(GR)_{80}$ flies, as coexpression of the viral antiapoptotic gene *p35* did not extend the lifespan of $(GR)_{80}$ flies (Fig. 2h).

Consistent with our findings in $(GR)_{80}$-expressing cortical neurons in mice[21], p62 inclusions and the p62 expression level were not increased in $(GR)_{80}$ flies (Supplementary Fig. 3)—further

evidence that poly(GR) expression alone does not cause p62 pathology, although p62-positive aggregates are common in postmortem *C9ORF72* patient brains[30–32] and in cells expressing poly(GA)[33,34]. To examine whether fly neurons expressing poly (GR) show abnormal localization or aggregation of TBPH (TAR DNA-binding protein 43 homolog), the *Drosophila* ortholog of human TDP-43, we generated a new antibody that is specific for endogenous TBPH, as shown by western blot analysis of wild-type and *TBPH* mutant flies (Supplementary Fig. 4). Moreover, immunostaining of 3-week-old $(GR)_{80}$ fly brains revealed no difference in TBPH aggregation in $(GR)_{80}$ and control flies (Fig. 2i). The lack of TBPH pathology, confirmed by western blot (Fig. 2j, k), is consistent with recent reports that poly(GR) expression in mouse neurons does not cause TDP-43 pathology[20,21]. Furthermore, ribosomal RNA biogenesis was not altered in $(GR)_{80}$ flies (Supplementary Fig. 5). Thus, poly(GR) is sufficient to induce locomotor defects, neurodegeneration, and premature death without causing several hallmarks of neuropathology found in postmortem *C9ORF72* patient brains.

**Lilliputian (lilli) strongly suppresses poly(GR) toxicity**. To further explore mechanisms of poly(GR) toxicity and discover potential therapeutic targets, we did an unbiased genetic screen with the Bloomington *Drosophila* Stock Center deficiency kit covering the second chromosome (Fig. 3a). Of 100 deficiency lines screened, 11 enhanced or decreased the survival of flies expressing $(GR)_{80}$ in all postmitotic neurons. Three of the strongest modifier deficiency lines, *Df(2L)C144*, *Df(2)ED4651*, and *Df(2)BSC180*, have an overlapping deleted genomic region (Supplementary Fig. 6), and each extended the lifespan of $(GR)_{80}$ flies (Fig. 3b). Only eight genes in the genomic region are covered by all three deficiencies (Supplementary Fig. 6a). Using genetic mutants and RNAi lines for these genes, we identified *lilliputian* (*lilli*) as a strong suppressor of $(GR)_{80}$ toxicity. For this experiment, three independent *lilli* mutant alleles were used; each resulted in loss of Lilli activity, shown by reduced expression levels of *lilli* mRNA or its target gene *Hsp70*[35] (Supplementary Fig. 6b, c). Interestingly, *lilli* was also one of the genes we identified as suppressors of poly(GR) toxicity in nonneuronal *Drosophila* cells[36]. Partial loss of Lilli activity induced by the three independent *lilli* mutant alleles did not affect the lifespan of control flies but extended the lifespan of poly(GR)-expressing flies by 10–20 days (Fig. 3c, d). These alleles also rescued the $(GR)_{80}$-induced climbing defect (Fig. 3e). However, partial loss of Lilli activity did not suppress the toxicity of FTD3-associated mutant CHMP2B (Supplementary Fig. 7), suggesting a certain degree of specificity. Overexpression of Lilli enhanced both the survival and climbing defects of poly(GR)-expressing flies (Fig. 3c, f), further confirming that *lilli* is a strong genetic modifier of poly(GR) toxicity.

**Lilli regulates transcription of long GC-rich sequences**. Lilli, the only *Drosophila* homolog of the human AF4/FMR2 family of transcription factors, is part of a SEC required for regulation of transcription elongation[37]. Therefore, we checked whether Lilli affects $(GR)_{80}$ mRNA levels. In our model, poly(GR) is encoded by $(GGXGCX)_{80}$, which is predicted to form G-quadruplex structures (by QGRS Mapper: http://bioinformatics.ramapo.edu/QGRS/analyze.php). Indeed, partial loss of Lilli activity, from crossing $(GR)_{80}$ flies with *lilli* genetic mutants, reduced $(GR)_{80}$ mRNA levels by 24–27% (Fig. 4a), whereas overexpression of Lilli increased the levels by twofold (Fig. 4b). Accordingly, poly(GR) protein levels in the brains of 2-week-old flies were reduced by partial loss of Lilli activity and increased by Lilli overexpression, as shown by immunostaining (Fig. 4c).

To further confirm this finding, we also used a highly sensitive ELISA, as we reported previously[21], to demonstrate the role of Lilli in regulating poly(GR) level (Fig. 4d). To examine whether partial loss of Lilli also rescues some downstream effects of poly (GR) toxicity, we first confirmed that poly(GR) expression increased the level of Ku80, a key protein in repair of double-stranded DNA breaks[38], in the heads of $(GR)_{80}$ flies as young as 1-week old (Supplementary Fig. 8a). Expression of Hsp70 and other heat shock proteins was also elevated in 1-week-old adult fly brains expressing $(GR)_{80}$ (Supplementary Fig. 8b), consistent with a recent report[39]. These molecular changes occurred 2 weeks before the death of these flies, suggesting they were early events in disease pathogenesis. The increases in the mRNA levels of *Ku80* and *Hsp70* (Fig. 4e, f) and of some other heat shock proteins (Supplementary Fig. 9) were attenuated by partial loss of Lilli activity, further indicating the beneficial effects of reduced Lilli activity in decreasing poly(GR) toxicity. $(GR)_{80}$ expression, on the other hand, did not alter *lilli* mRNA levels (Supplementary Fig. 10).

To see whether Lilli specifically regulates the $(GR)_{80}$ level, we expressed another transgene, *UAS-GFP*, with the *ELAV-GAL4* driver and measured *GFP* mRNA levels by quantitative PCR (qPCR). Partial loss of Lilli did not alter *GFP* mRNA levels (Supplementary Fig. 11a) but did decrease the mRNA level of *CONT-(GR)$_{80}$*, which has the same $(GGXGCX)_{80}$ DNA sequence as $(GR)_{80}$ flies except that the start codon is replaced with a stop codon (Supplementary Fig. 11b). These results demonstrate the specificity of Lilli function in flies.

To ascertain whether Lilli also regulates the expression of $G_4C_2$ repeat RNA, we used a fly model we generated previously in which 5 or 160 copies of $G_4C_2$ repeats are flanked by human intronic and exonic sequences (Fig. 5a)[40]. Partial loss of Lilli activity decreased the mRNA level of the *C9ORF72* minigene containing 160 $G_4C_2$ repeats by 20–30% (Fig. 5b), but did not affect the mRNA level of the *C9ORF72* minigene containing only five $G_4C_2$ repeats (Fig. 5c). Thus, Lilli seems to preferentially regulate the transcription of expanded $G_4C_2$ repeat sequences. Expression of intronic 160 copies of $G_4C_2$ repeats in *Drosophila* motor neurons resulted in numerous nuclear RNA foci (Fig. 5d), as we reported previously[40]. Reducing Lilli function with either a *lilli* mutant allele or a *lilli* RNAi significantly decreased the number of RNA foci per cell (Fig. 5e), consistent with decreased transcription of the *C9ORF72* minigene containing 160 $G_4C_2$ repeats.

**Effect of AFFs on *C9ORF72* levels in patient cortical neurons**. There are four mammalian homologues of the *Drosophila* Lilli protein: AFF1/AF4, AFF2/FMR2, AFF3/LAF4, and AFF4/AF5q31 (Fig. 6a), which constitute the AFF (AF4/FMR2) family of proteins. AFF1 and AFF4 are part of SEC required to release polymerase II (Pol-II) from pausing sites to start productive elongation of most genes. AFF2 and AFF3 are key subunits of the SEC-like complexes SEC-L2 and SEC-L3, respectively, which are required for transcription elongation of only a small subset of genes[37]. *AFF2* is also known as *FMR2* (fragile mental retardation 2); loss of its expression due to a CCG expansion in the 5′UTR results in a mild to borderline intellectual disability, fragile XE syndrome[41,42].

To identify the AFF family members required to produce expanded $G_4C_2$ repeat RNA in *C9ORF72* patient neurons, we obtained 3–4 short hairpin RNAs (shRNAs, Dharmacon) targeting each of the four family members and tested their knockdown efficiency in HEK293 cells. The two best-performing shRNAs for each AFF family protein were transduced into *C9ORF72* cortical neurons differentiated from iPSC lines derived

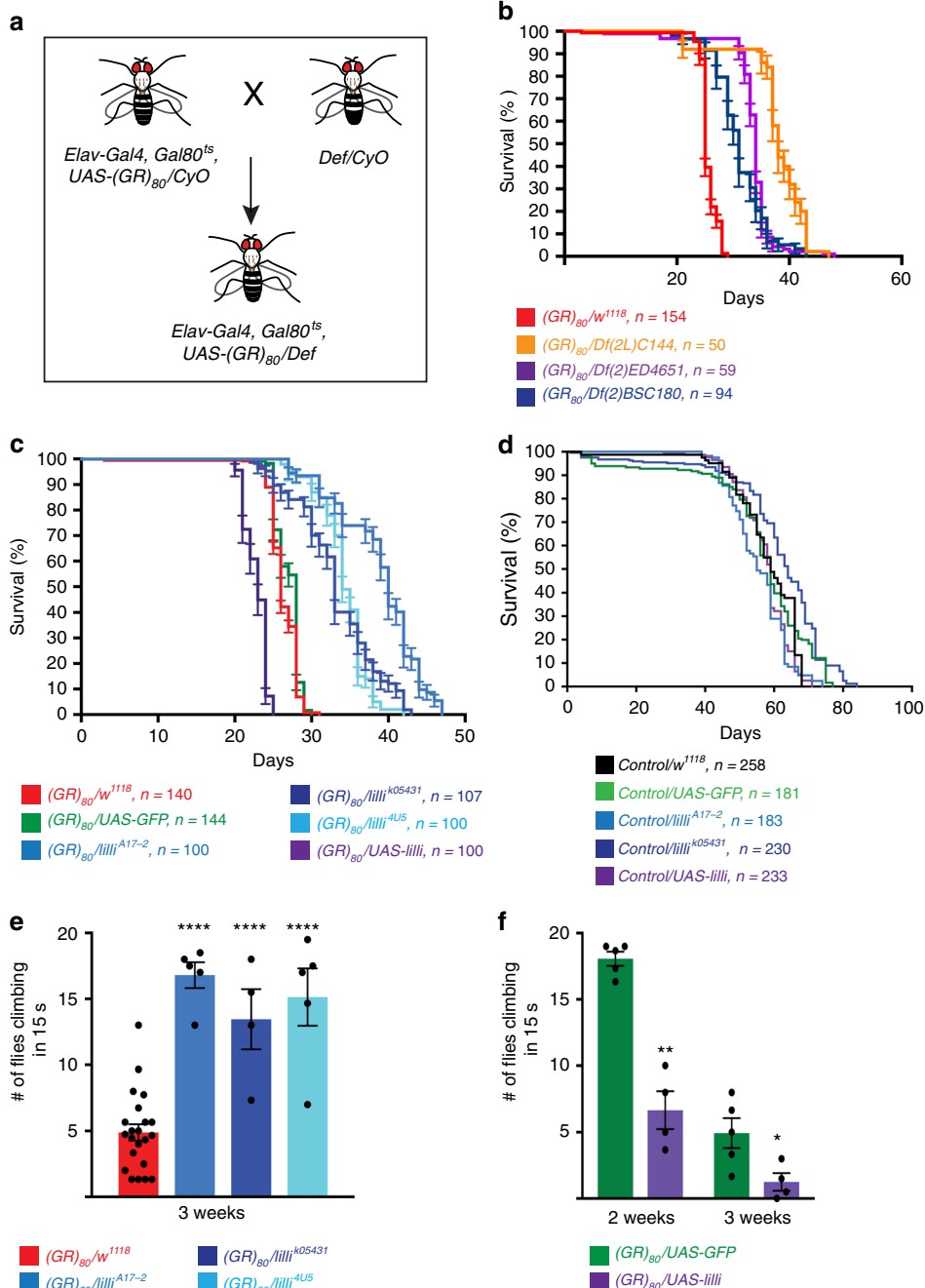

**Fig. 3** Identification of *lilli* as a strong genetic suppressor of poly(GR) toxicity. **a** The design of a genetic deficiency screen in *Drosophila*. Def, deficiency lines. **b** Deficiencies Df(2L)C144, Df(2)ED4651, and Df(2)BSC180 in trans significantly extended the lifespan of $(GR)_{80}$ flies. The n number for each genotype is in the figure. $p < 0.0001$ (log-rank test). **c** Survival analysis of $(GR)_{80}$ flies that are heterozygous for different *lilli* mutant alleles. Partial loss of Lilli activity by $lilli^{A17-2}$, $lilli^{4U5}$, and $lilli^{k05431}$ heterozygosity extended median survival by 14, 8, and 7 days, respectively. $p < 0.0001$ (log-rank test). Lilli overexpression decreased the median survival by 3 days. $p < 0.0001$ (log-rank test). The n number for each genotype is in the figure. **d** Effect of partial loss of Lilli activity on lifespan of control flies. The n number for each genotype is in the figure. **e** Reducing Lilli function rescues the climbing defect of $(GR)_{80}$ flies in the negative geotaxis assay. One dot = 20 flies, $n = 22, 5, 4$, and 5 assays for genotypes from the left to the right, respectively. $****p < 0.0001$ (one-way ANOVA, Dunnett's test). **f** Overexpression of Lilli enhanced the climbing defect of $(GR)_{80}$ flies. One dot = 20 flies, $n = 5, 4, 5, 4$ assays for genotypes from the left to the right, respectively. $*p < 0.05$; $**p < 0.01$ (Welch's test). Source data are provided as a Source Data file.

from two *C9ORF72* patients. About 80% of neurons generated with this protocol are MAP2-positive[43] (Fig. 6b). Four-week-old neurons were transduced with lentiviruses expressing the shRNAs targeting each AFF family member and cultured for 10 days. *AFF2* and *AFF3* shRNAs reduced target gene expression by 60–70% (Fig. 6c, d). Under these conditions, levels of *C9ORF72* variant 3 (V3) mRNA, which encodes full-length C9ORF72

protein and whose precursor mRNA contains the expanded $G_4C_2$ repeats in the first intron, decreased by 15% after *AFF2* knockdown (Fig. 6c) but were unaffected by *AFF3* knockdown (Fig. 6d). Gene expression changes after *AFF1* or *AFF4* knockdown could not be measured, as all reference genes tested were also affected (Supplementary Fig. 12). This observation is consistent with the participation of AFF1 and AFF4 in the

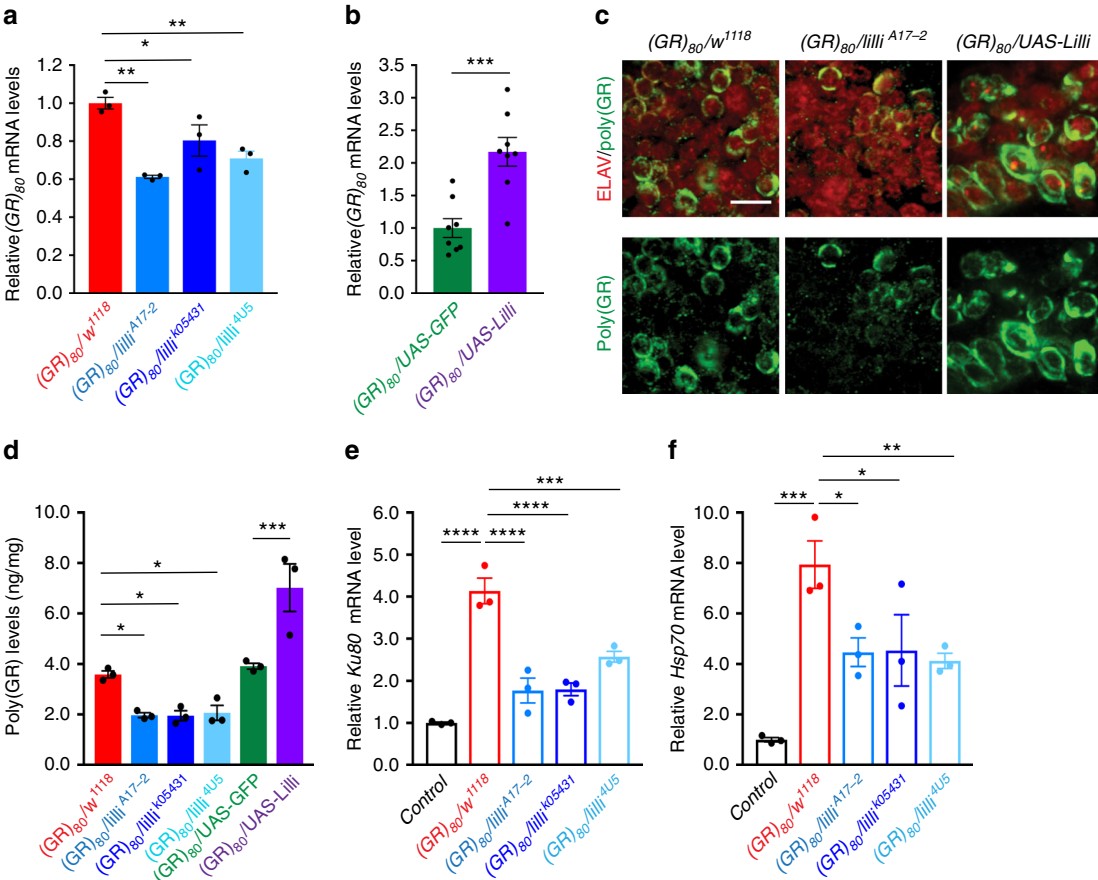

**Fig. 4** Partial loss of Lilli suppresses poly(GR) toxicity. **a**, **b** QPCR analysis of $(GR)_{80}$ mRNA level in 2-week-old fly heads. $n = 3$ independent crosses in **a** and $n = 8$ independent crosses in **b**. Values are mean ± s.e.m. *$p < 0.05$; **$p < 0.01$, ***$p < 0.001$ (one-way ANOVA, Dunnett's test (**a**), Welch's $t$ test (**b**)). **c** Immunostaining of 2-week-old fly brains for ELAV (red) and poly(GR) (green). Scale bar: 5 μm. **d** Poly(GR) levels in lysates of heads from 2-week-old flies were determined by MSD immunoassay ($n = 3$ independent crosses). Values are mean ± s.e.m. *$p < 0.05$, ***$p < 0.0001$ (one-way ANOVA, Dunnett's test). The expression levels of Ku80 (**e**) and Hsp70 (**f**) mRNAs in 2-week-old fly heads were assayed by qPCR. Values are mean ± s.e.m. $n = 3$ independent crosses. *$p < 0.05$; **$p < 0.01$, ***$p < 0.001$, ****$p < 0.0001$ (one-way ANOVA, Dunnett's test). Source data are provided as a Source Data file.

transcriptional regulation of the expression of most genes[44]. Thus, AFF2 may have a unique function in the production of expanded $G_4C_2$ repeat RNA in patient neurons.

**Isoform- and allele-specific regulation of C9ORF72 by AFF2.** To further investigate how AFF2 regulates C9ORF72 expression in human neurons, we used CRISPR-Cas9 technology to knock-out AFF2 in iPSC lines from two C9ORF72 patients. The genomic editing experiments were designed to delete in exon 8 a region common to all eight reported AFF2 isoforms, thereby creating a frameshift and a premature stop codon (Fig. 7a). Because AFF2 is located on chromosome X, cells with only 50% loss of AFF2 cannot be obtained due to X-inactivation in female cells or presence of only one X chromosome in male cells. To simplify clonal selection after CRISPR-Cas9 deletion of AFF2 in the first parental iPSC line (female karyotype), we selected lines that were homozygous for the deletion to avoid obtaining lines with a heterozygous deletion in the inactive X chromosome. A total of five lines were selected across the two patients; two of these lines, from parental iPSCs of one patient, had an 88 base-pair (bp) deletion as well as a 2-bp insertion in exon 8; three lines, from parental iPSCs of the second patient, had a 94- or 109-bp deletion in exon 8 (Fig. 7b and Supplementary Fig. 13). qPCR analysis of AFF2 mRNA levels in iPSCs confirmed the success of the CRISPR-Cas9-mediated knockout (Fig. 7c). Then, we tested the

effect of AFF2 knockout on the transcription of different C9ORF72 variants with or without repeats in sense or antisense directions. As a consequence of AFF2 knockout, C9ORF72-V3 mRNA levels were also significantly lower in iPSC lines than in the respective parental line, while C9ORF72-V2 (which encodes the full-length C9ORF72 protein but whose precursor mRNA does not contain the expanded $G_4C_2$ repeats) and first intron antisense levels were unchanged (Supplementary Fig. 14). After differentiation of the iPSCs into cortical neurons, AFF2 expression was still much lower in the knockout lines than in the parental line (Fig. 7d). As in iPSCs, AFF2 knockout significantly reduced the level of C9ORF72-V3, but not of isoform V2 or antisense RNA (Fig. 7e–g).

Next, we analyzed allele-specific expression of the junction between exon 1b and exon 2 in C9ORF72-V3 using pyrosequencing, taking advantage of the presence of the rare single-nucleotide polymorphism (SNP; rs10757668) in exon 2 in the parental and respective AFF2 knockout iPSC lines. Because the expanded $G_4C_2$ repeats are associated with the G-allele[6,45], we could determine how AFF2 knockout affects each allele (Fig. 7h). Relative expression of the V3-specific G-allele was much lower in AFF2 knockout lines than in the respective parental lines (Fig. 7i). Together, these results suggest that AFF2 preferentially regulates the transcription of the C9ORF72 allele containing expanded $G_4C_2$ repeats in patient neurons.

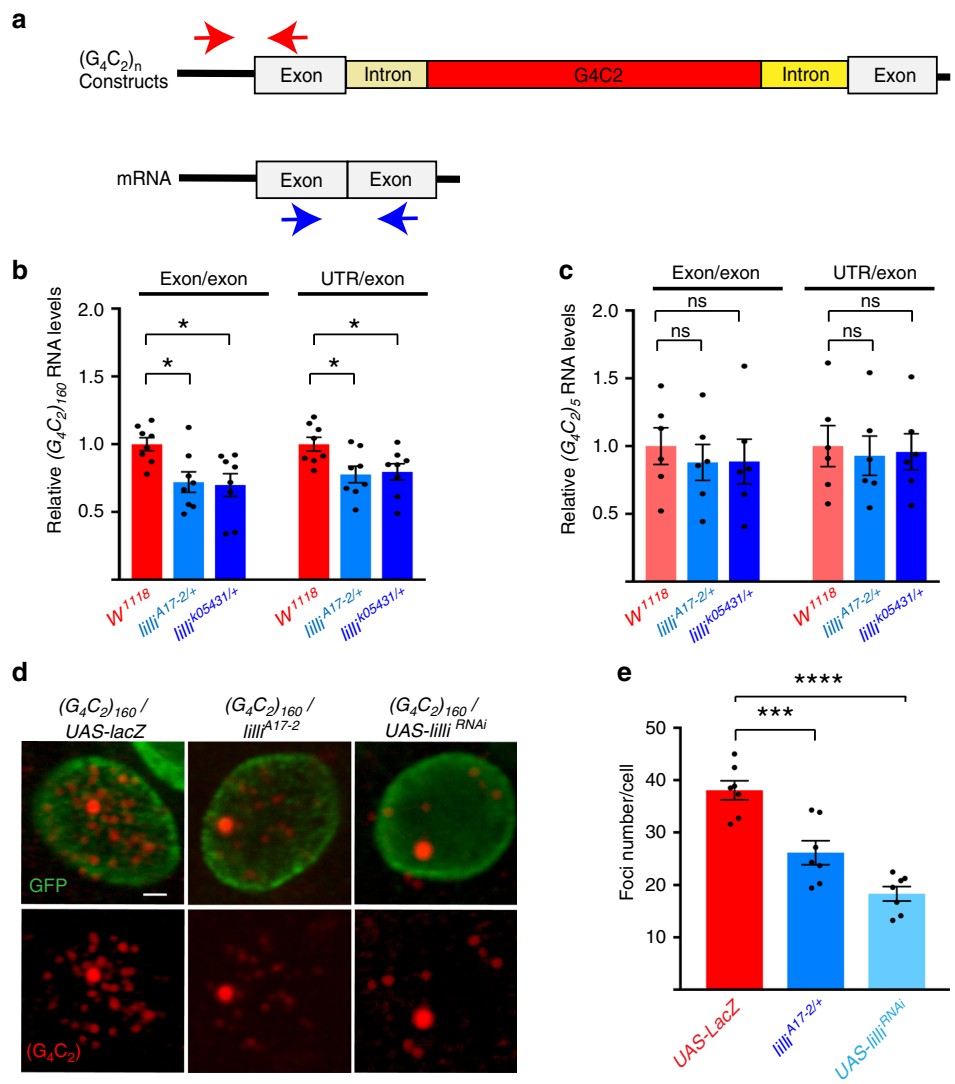

**Fig. 5** Lilli regulates expression of expanded $G_4C_2$ repeat RNA. **a** Schematic of the intronic $G_4C_2$ repeat model and locations of the primers used for qPCR (red and blue arrows). **b** qPCR analysis of $(G_4C_2)_{160}$ repeat RNA levels in the *Drosophila* eye. Values are mean ± s.e.m. *$p < 0.05$ (one-way ANOVA, Dunnett's test). **c** qPCR analysis of $(G_4C_2)_5$ repeat RNA levels in the *Drosophila* eye. Values are mean ± s.e.m. ns not significant (one-way ANOVA, Dunnett's test). $n = 8$ independent crosses in **b** and $n = 6$ in **c**. **d** Formation of nuclear RNA foci by $(G_4C_2)_{160}$ repeat RNA in *Drosophila* motor neurons. Red, RNA foci; green, nuclear GFP. Scale bar: 0.5 μm. **e** Number of RNA foci in motor neurons expressing $(G_4C_2)_{160}$ with or without partial loss of Lilli activity. $n = 7$ animals per genotype. In each animal, 5–7 motor neurons were examined. ***$p < 0.001$, ****$p < 0.0001$ (one-way ANOVA, Dunnett's test). Source data are provided as a Source Data file.

***AFF2* knockout reduces RNA foci and DPRs in human neurons.** Accumulation of nuclear RNA foci and production of DPR proteins are consequences of the expression of expanded $G_4C_2$ repeats. Using in situ hybridization, we found that the percentage of iPSCs containing RNA foci, and the average number of foci per cell, were significantly lower in *AFF2* knockout lines than in the respective parental lines (Supplementary Fig. 15). Consistently, *AFF2* knockout also reduced the number of RNA foci in *C9ORF72* iPSC-derived cortical neurons (Fig. 8a–c). Poly(GR) and poly(GP) protein levels were also decreased in *C9ORF72* iPSCs and *C9ORF72* iPSC-derived cortical neurons after *AFF2* knockout (Fig. 8d, e and Supplementary Fig. 16). *AFF2* expression was similar in *C9ORF72* patient brain tissues and control tissues (Supplementary Fig. 17), suggesting that *AFF2* is not upregulated in patient brains.

**Rescue of disease phenotypes by *AFF2* loss in human neurons.** To examine whether TDP-43 pathology such as cytoplasmic translocation can be detected in *C9ORF72* patient iPSC-derived neurons, we used the neurotrophic factor withdrawal approach. In the following set of experiments, in addition to the *C9ORF72* parental and *AFF2* knockout lines, we also used an isogenic no repeat line generated recently[36]. After 2–3 weeks without neurotrophic factors, a higher percentage of *C9ORF72* neurons than the respective isogenic no repeats controls had cytoplasmic TDP-43 (Fig. 9a–c). This phenotype was relatively mild as most of the TDP-43 remained in the nucleus and cytoplasmic inclusions were only occasionally observed (Fig. 9a). Interestingly, *AFF2* knockout prevented translocation of TDP-43 to the cytoplasm. When the neurons were kept in medium containing all the neurotrophic factors, there was no difference between the different groups of samples (Supplementary Fig. 18).

Since our $(GR)_{80}$ flies showed signs of axonal degeneration, we looked for this phenotype in *C9ORF72* patient-derived cortical neurons. Indeed, 2–3 weeks of neurotrophic factor withdrawal increased axonal degeneration in *C9ORF72* neurons, and this

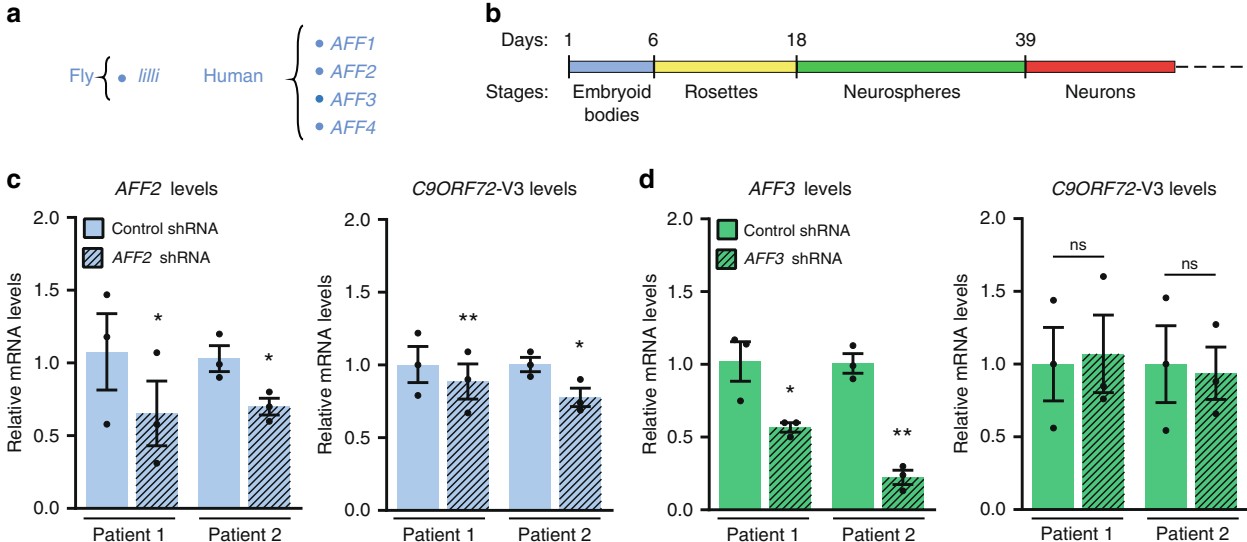

**Fig. 6** The effect of knockdown of AFF family members on *C9ORF72* expression in human neurons. **a** Corresponding human homologues of *Drosophila lilli*. **b** Schematic representation of the neuron differentiation protocol used in this study. Human neurons derived from two *C9ORF72* patient iPSC lines were transduced with lentiviruses containing *AFF2*-specific (**c**) or *AFF3*-specific (**d**) shRNAs, cultured for 10 days, and analyzed for expression of *AFF2*, *AFF3*, and *C9ORF72*-V3 mRNAs. Each data point represents one differentiation ($n = 3$ independent differentiations). Values are mean ± s.e.m. ns, not significant. *$p < 0.05$; **$p < 0.01$ (paired two-tailed Student's *t* test). Source data are provided as a Source Data file.

phenotype was prevented by *AFF2* knockout (Fig. 9d–f). Both the increases in TDP-43 cytoplasmic localization and the axonal degeneration were replicated in neurons derived from a second *C9ORF72* patient and respective isogenic control (Supplementary Fig. 19).

## Discussion

In this study, we established a *Drosophila* poly(GR) model in which expression of 80 repeats of GR in adult neurons causes axonal degeneration, shortened lifespan, and locomotion defects without significant cell death or TDP-43 pathology. In an unbiased deficiency screen in this model, the *lilli* gene, encoding a component of the SEC, was identified as a novel strong modifier of poly(GR) toxicity. Because in this model, poly(GR) was expressed in adult neurons and lifespan was used as the readout for the genetic screen, it is possible to identify novel modifier genes not seen in other screens. Downregulation of *lilli* reduced poly(GR) RNA and protein levels. Lilli also reduced the levels of RNA containing long $G_4C_2$ repeat sequences without affecting the expression of short repeat sequences. Moreover, CRISPR-Cas9-mediated loss of AFF2/FMR2, one of four human homologs of Lilli, specifically decreased the levels of *C9ORF72* RNA, RNA foci, and DPR proteins in neurons derived from iPSCs of patients with *C9ORF72*-ALS/FTD. These findings uncover a role for AFF2/FMR2 in controlling the expression and toxicity of *C9ORF72* alleles containing expanded $G_4C_2$ repeats in ALS/FTD.

A series of recent findings indicate that poly(GR) has pleiotropic effects and compromises several molecular pathways[1,2]. Our findings here suggest that axonal degeneration is another potential pathogenic mechanism in *C9ORF72*-ALS/FTD. During normal aging and in various neurodegenerative diseases, an altered stress response could lead to axonal defects[46]. Indeed, poly(GR) increases the levels of oxidative stress[19] and heat shock response genes[39] (Fig. 4e, f). Moreover, poly(GR)-induced axonal degeneration precedes neuronal cell loss, which may account for neuronal functional deficits during disease progression, as in other models of neurodegenerative diseases[46,47]. This important observation in our fly model is supported by the finding of axonal

degeneration in *C9ORF72* patient-derived neurons. Nevertheless, it remains to be determined how poly(GR) causes axonal degeneration at the molecular level. On the other hand, the lack of TDP-43 pathology in this fly model is consistent with observations that poly(GR) expression in mice does not induce TDP-43 pathology[20,21]. Thus, this key pathological hallmark of *C9ORF72*-ALS/FTD may be caused by other DPR proteins or a combination of DPR proteins and/or toxic repeat RNA. Indeed, we found evidence for a mild form of TDP-43 pathology in *C9ORF72* patient-derived neurons.

In our unbiased genetic screen in *Drosophila*, a 50% reduction of Lilli function was sufficient to decrease poly(GR) levels and attenuate its toxicity. The effect of partial loss of Lilli activity seems to be relatively specific to the $(GGXGCX)_{80}$ sequence, as the levels of other transgenes expressed with the same GAL4 driver were unaffected. Interestingly, loss of Lilli function also suppressed the toxicity of TDP-43 without affecting its level in *Drosophila*[48]. Moreover, partial loss of Lilli function affected the expression of $(G_4C_2)_{160}$ but not $(G_4C_2)_5$, suggesting that the transcription of expanded $G_4C_2$ repeats in *Drosophila* is more sensitive to Lilli function. Findings in *Drosophila* models of neurodegeneration often need to be confirmed in patient iPSC-derived neurons[49]. Indeed, CRISPR-Cas9-mediated *AFF2* knockout in *C9ORF72* iPSC-derived patient neurons suggests that AFF2 preferentially regulates the transcription of the *C9ORF72* allele containing expanded $G_4C_2$ repeats. AFF2 is a subunit of the SEC-like complex SEC-L2, which is required for transcription elongation of only a small subset of genes[44]. Thus, transcription elongation can be a rate-limiting step for the transcription of the long GC-rich sequences, likely in part because the high GC content favors the formation of R-loops that contribute to Pol-II pausing[50].

Although complete loss of AFF2 function causes mild intellectual disability[41,42], partial knockdown of AFF2 may still represent a potential therapeutic approach since *AFF2* knockout in *C9ORF72* iPSC-derived patient neurons largely rescued axonal degeneration and TDP-43 pathology. Moreover, our findings here highlight the need to further investigate other components of transcription elongation complexes as potential therapeutic

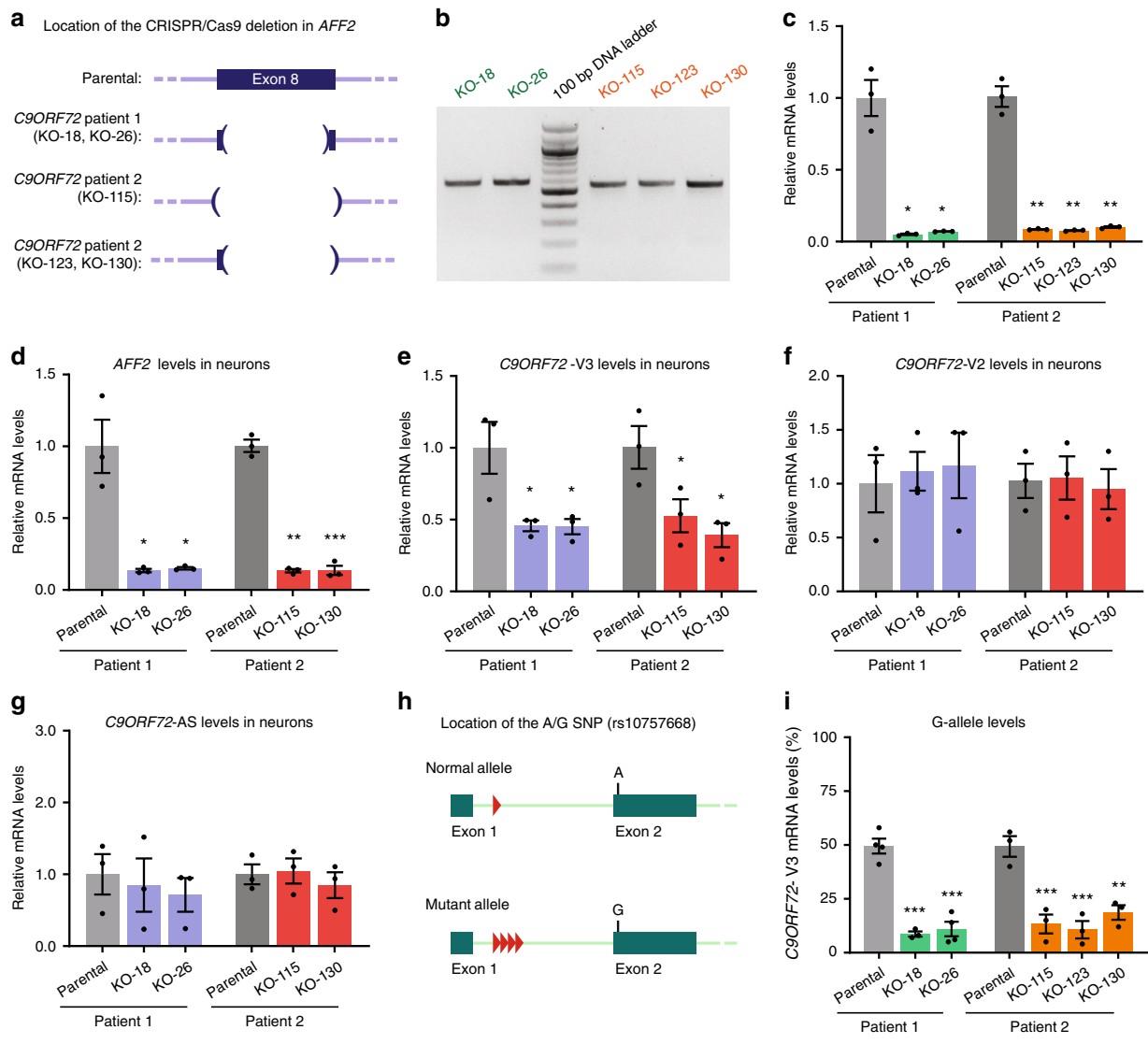

**Fig. 7** AFF2 regulates expression of expanded $G_4C_2$ repeats. **a** Schematic of the CRISPR-Cas9-mediated deletions created in exon 8 of *AFF2* to generate *AFF2* knockout (KO) *C9ORF72* iPSC lines. **b** PCR amplification of the genomic region containing exon 8, showing that all lines were homozygous for the deletion. **c** *AFF2* mRNA levels in parental and *AFF2*-KO iPSC lines. Neurons from CRISPR-Cas9-edited lines were analyzed for expression of *AFF2* (**d**), *C9ORF72*-V3 (**e**), *C9ORF72*-V2 (**f**), and *C9ORF72* antisense (**g**) mRNAs. **h** Schematic of the location of the A/G SNP (rs10757668) in exon 2 of *C9ORF72*. **i** Pyrosequencing quantification of the relative expression levels of the allele containing expanded $G_4C_2$ repeats (G-allele) in parental and *AFF2*-KO iPSC lines. Each data point represents one differentiation ($n = 3$ independent differentiations). Values are mean ± s.e.m. *$p < 0.05$; **$p < 0.01$; ***$p < 0.001$ (**c**, **d**, Welch's $t$ test; **e**, **i**, one-way ANOVA, Dunnett's multiple comparisons test). Source data are provided as a Source Data file.

targets. Several factors may be needed to facilitate RNA Pol-II processivity on long GC-rich sequences. For example, PAF1 complex, a positive regulator of RNA Pol-II transcription elongation, has recently been identified as a modifier of $G_4C_2$ repeat toxicity that regulates the transcription of expanded repeat sequence in *Drosophila*[51]. In addition, SUPT4H1 is also required for the transcription of expanded $G_4C_2$ repeats, and partial knockdown of SUPT4H1 in *C9ORF72* iPSC-derived patient neurons decreases the levels of $G_4C_2$ repeat RNA and DPR proteins[52]. SUPT4H1 is a component of DSIF (DRB sensitivity-inducing factor) that associates with negative elongation factor (NELF) to promote promoter-proximal pausing of RNA Pol-II[53]. Transcription elongation resumes when a component of SEC, positive elongation factor b, releases the pause by phosphorylating RNA Pol-II, NELF, and DSIF[37]. Although lowering the level of *SUPT4H1* by ~90% reduces global RNA levels[54], other factors in the transcription elongation regulatory complexes such as AFF2 may more specifically influence the expression of expanded

$G_4C_2$ repeats, and therefore are potential therapeutic targets worthy of further investigation.

## Methods

***Drosophila* stocks and genetics.** All flies were raised on standard food at 25 °C. *UAS-(GR)$_{80}$* and *UAS-CONT-(GR)$_{80}$* flies as well as *UAS-(G$_4$C$_2$)$_{160}$* and *UAS-(G$_4$C$_2$)$_5$* flies have been obtained from earlier studies[40]. A deficiency kit for the second chromosome and *ELAV-GAL4*, *TubGAL80$^{ts}$*, *UAS-p35*, *lilli$^{k05431}$*, *lilli$^{A17-2}$*, *UAS-GFP*, *UAS-lacZ*, *OK$^{371}$-GAL4*, and *GMR-GAL4* fly stocks were from the Bloomington Drosophila Stock Center. *Lilli* RNAi line (v106142) was from the Vienna Drosophila Resource Center (VDRC). *lilli$^{4U5}$* and *UAS-lilli-HA* fly stocks were a gift from Dr Ernst Hafen. The *lilli$^{A17-2}$* allele was created by ethyl metha-nesulfonate mutagenesis. The nature of the mutation is unknown but is thought to be a loss of function allele[55]. The *lilli$^{k05431}$* allele is a P-element insertion[56]. The *lilli$^{4U5}$* allele has a short deletion resulting in a frameshift and a functional null allele[56].

**Lifespan assay.** All fly crosses were kept at 18 °C until eclosion. Male flies were collected 24 h after eclosion and maintained at 25 °C. For survival analysis,

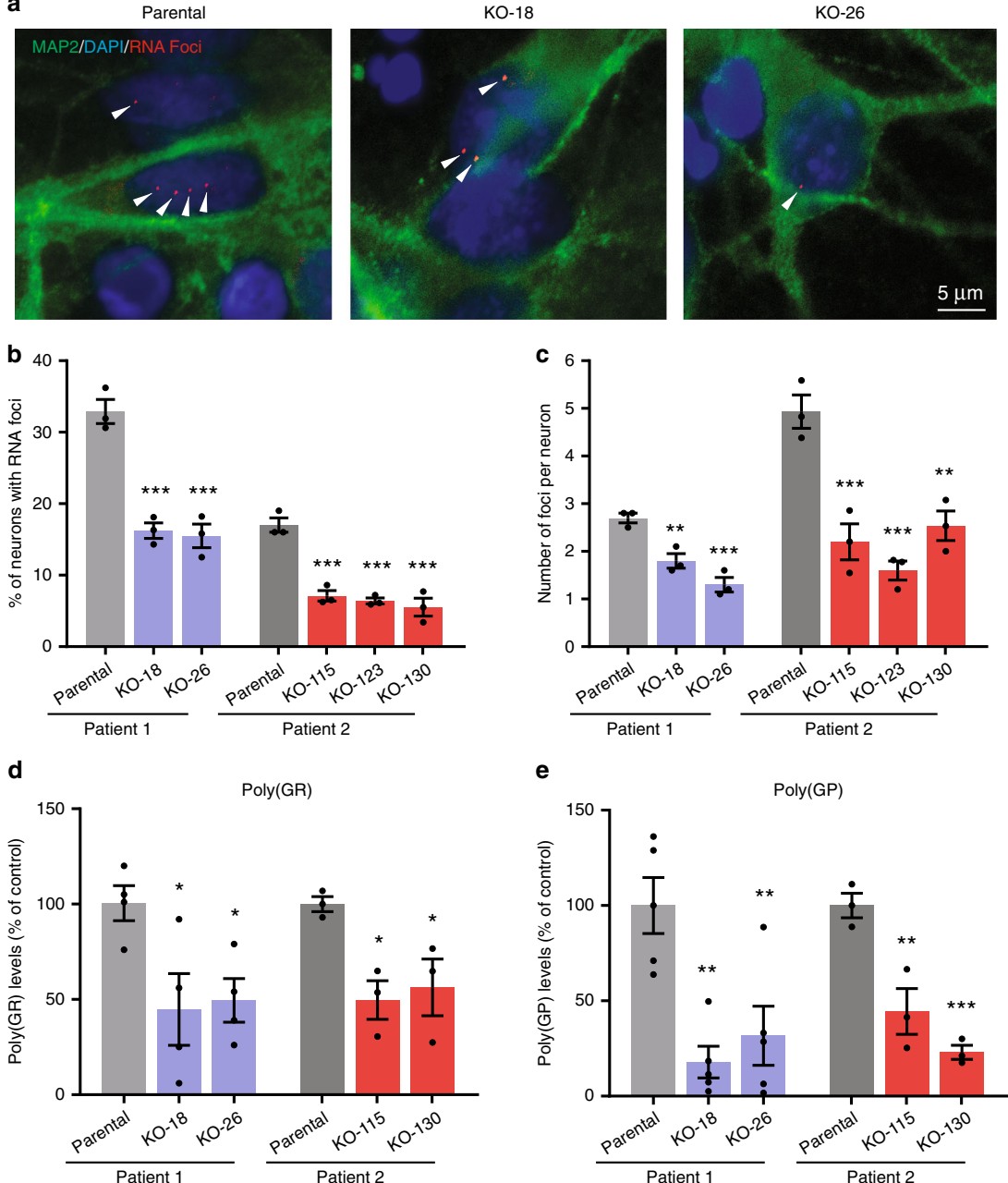

**Fig. 8** *AFF2* knockout (KO) reduces RNA foci and DPR proteins. **a** Representative images of fluorescence in situ hybridization (FISH) analysis of parental and respective *AFF2*-KO neurons with a Cy3-conjugated $(G_2C_4)_4$ probe. RNA foci are red, neurons are green (MAP2), and the nuclei are blue (DAPI). **b** Percentage of *C9ORF72* iPSC-derived neurons containing RNA foci labeled with a Cy3-conjugated $(G_2C_4)_4$ probe. **c** Average number of RNA foci per neuron. Only neurons containing RNA foci were counted. Each data point is the average number of RNA foci in at least 100 neurons from an independent differentiation. Poly(GR) (**d**) and poly(GP) (**e**) levels in *AFF2*-KO *C9ORF72* neurons measured by MSD immunoassay. Each data point represents one differentiation ($n = 3$ independent differentiations for all panels except $n = 4$ independent differentiations for patient 1 in **d** and $n = 5$ independent differentiations for patient 1 in **e**). Values are mean ± s.e.m. *$p < 0.05$; **$p < 0.01$; ***$p < 0.001$ (one-way ANOVA, Dunnett's multiple comparisons test). Source data are provided as a Source Data file.

100–200 flies of each genotype were used. Flies were transferred to fresh food every other day, and dead flies were scored.

**Climbing assay**. Twenty male flies were placed in a 10-cm-long plastic vial and tapped to the bottom of the vial, and the number of flies that climbed 5 cm in 15 s was counted. For consistency, all flies were tested at the same time of the day. For each genotype, 80–100 flies were tested.

**Neuronal cultures and human brain samples**. Published iPSC lines from two *C9ORF72* carriers (one with a female and the other with male karyotype)

heterozygous for the A/G SNP (rs10757668) in exon 2 of *C9ORF72* were differentiated into cortical neurons following a protocol established in the laboratory[43]. Briefly, iPSC colonies were lifted after incubation with 1:2 Accutase/PBS and grown as embryoid bodies (EBs) in suspension for 4 days in basic fibroblast growth factor-free iPSC medium containing dorsomorphin and A-83. EBs were cultured for 2 additional days in neuronal induction medium supplemented with cyclopamine and then allowed to attach and form neural tube-like rosettes. Fifteen-day-old rosettes were collected and grown in suspension as neurospheres. Neurospheres were dissociated after 3 weeks, and the cells were seeded on plates coated with poly-D-lysine and laminin. Neurons were cultured for 4 weeks, and transduced with the lentiviral particles (multiplicity of infection, 100) for 10 additional days. All

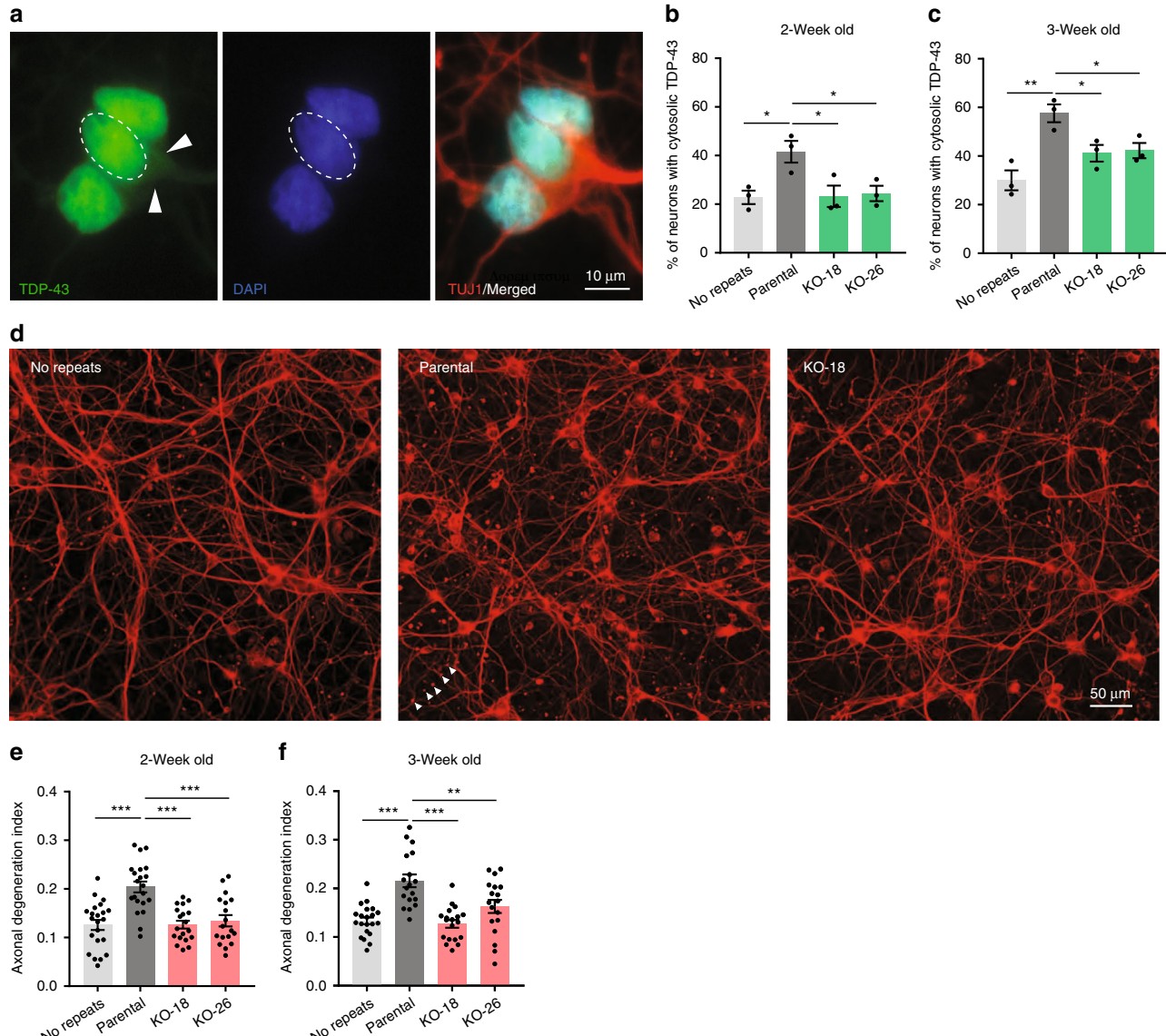

**Fig. 9** *AFF2* knockout rescues disease phenotypes. **a** Representative immunofluorescence image of the TDP-43 localization in cortical neurons. Arrowheads indicate TDP-43 in the cytoplasm. Quantification of the percentage of *C9ORF72* iPSC-derived neurons showing cytoplasmic TDP-43 after neurotrophic factor withdrawal for 2 weeks (**b**) or 3 weeks (**c**). **d** Axonal degeneration was examined with the use of the marker βIII-tubulin (TUJ1), which allows the visualization of swollen varicosities and axonal fragments. Panels show representative immunofluorescence images for no repeats, parental and *AFF2*-KO-18 neuronal cultures. Arrowheads show an example of a fragmented axon. Quantification of axonal degeneration by measuring the ratio of fragmented axons over the total TUJ1$^+$ area 2 weeks (**e**) or 3 weeks (**f**) after neurotrophic factor withdrawal. Six to eight randomly selected fields were analyzed for each condition and each neuronal culture. Each independent data point represents one field. $n = 3$ independent differentiations. Values are mean ± s.e.m. *$p < 0.05$; **$p < 0.01$; ***$p < 0.001$ (one-way ANOVA, Dunnett's multiple comparisons test). Source data are provided as a Source Data file.

experiments were done with neurons derived from 3–5 independent differentiations. Human brain cDNA samples were from a previous study[40]. The mean age at death was 78 ± 8 years in seven control subjects and 65 ± 7 years in six *C9ORF72* repeat expansion carriers.

A shorter cortical differentiation protocol was used for the TDP-43 and axonal degeneration assays. Briefly, iPSC colonies were seeded on Matrigel-coated wells in mTeSR1 medium (StemCell Technologies); 24 h later, the medium was changed to neuroepithelial progenitor (NEP) medium (1:1 DMEM/F12:neurobasal, 0.5× N2, 0.5× B27, 0.1 mM ascorbic acid (Sigma), 1× Glutamax, 3 μM CHIR99021 (Tocris Bioscience), 2 μM DMH1 (Tocris Bioscience), and 2 μM SB431542 (Stemgent)) and replaced every other day for 6 days. Progenitor colonies were dissociated with Accutase, seeded at 1:6 on Matrigel-coated wells, and cultured in NEP medium containing 2 μM cyclopamine (StemCell Technologies) for 6 days; the medium was replaced every other day. Cortical progenitors were lifted, cultured in suspension for 6 additional days in the absence of CHIR99021, DMH1, and SB431542, and dissociated to single cells with Accutase. Cells were seeded on poly-lysine/laminin-coated coverslips (Corning) in neurotrophic factor withdrawal medium (1:1 DMEM/F12:neurobasal, 0.5× N2, 0.5× B27, 0.1 mM ascorbic acid (Sigma),

1× Glutamax, 0.1 μM compound E (Calbiochem), and 1 μg/ml laminin (Sigma)) for 2–3 weeks. This protocol generated a culture with >90% TUJ1$^+$ neurons. The work involving iPSCs was approved by the Institutional Biosafety Committee of the University of Massachusetts Medical School, Worcester.

**Lentivirus production.** shRNAs were obtained from Dharmacon (clone IDs TRCN0000119090 and TRCN0000119091 for *AFF2* and TRCN0000108066 and TRCN0000108068 for *AFF3*). High-titer lentiviral preparations were produced by the Viral Vector Core at the University of Massachusetts Medical School.

**Generation of *AFF2*-KO iPSC lines.** Genome editing with the CRISPR-Cas9 system to create a deletion to knock out *AFF2* was done by ALSTEM (Richmond, CA). Briefly, the Neon electroporation system was used to transfect iPSCs with two different guide RNAs: ATCTCTTTGTAGGATGCTTG and TCTAACCTGGTAG AGCTCAG for patient 1 iPSCs and ATCTCTTTGTAGGATGCTTG and AGCC CTAAGCTCTTCTAACC for patient 2 iPSCs. Single cells were placed in 96-well plates, cultured for 14 days, expanded, and cultured in 24-well plates. Genomic

DNA from each clone was extracted with the Zymo genomic extraction kit. Clones with the desired homozygous deletion were identified with a PCR amplification assay, and the PCR products were sent for sequencing.

After we received the frozen vials containing each clone from ALSTEM, iPSCs were recovered, expanded, and collected to isolate genomic DNA. The region of interest was amplified by PCR, and the products were sent for sequencing to confirm the identity of each clone.

**RNA extraction and quantitative PCR.** For flies, total RNA was extracted from adult heads with the Qiagen miRNeasy kit (Cat. No. 217004) following the manufacturer's protocol and reverse transcribed with random hexamers and SuperScript II Reverse Transcriptase (Thermo Fisher Scientific, Cat. No. 18064014). For iPSCs, neurons, and brain tissue, total RNA was extracted with the Qiagen RNeasy kit (Cat. No. 74104) and treated with DNase I (Qiagen, Cat. No. 79254). RNA (500–900 ng) was reverse transcribed into cDNA with random hexamers or *C9ORF72* antisense-specific reverse primer and MultiScribe reverse transcriptase (Thermo Fisher Scientific, Cat. No. N8080234) according to the manufacturer's instructions. qPCR was done with an Applied Biosystems Quant Studio 3 system and SYBR Select Master Mix (Thermo Fisher Scientific, Cat. No. 4472908). Ct values for each sample and gene were normalized to cyclophilin. The relative expression of each target gene was determined with $2^{-\Delta\Delta Ct}$ method. Primers used for qPCR are listed in Supplementary Table 1. In the *AFF1* and *AFF4* knockdown experiments, the following reference genes were tested: *ACTN*, *GAPDH*, *HPRT*, and *SUPT5H* (as suggested by Genevestigator software: https://genevestigator.com/gv/).

**Quantification of *C9ORF72* allele-specific expression.** Total RNA (1 μg) from parental and respective knockout iPSCs was used to generate cDNA with the OneStep RT-PCR Master Mix (Qiagen, Cat. No. 978801) and V3-specific primers (5′-GAGCAGGTGTGGGTTTAGGAGA-3′ and 5′-TCACTGCATTCCAACTGTC ACAT-3′). Pyrosequencing analysis was done on a PyroMark Q24 instrument with the sequencing primer (5′-TTCCAACTGTCACATTATC-3′) according to the manufacturer's instructions. The results are expressed as the percentage of V3 mRNA containing the G-allele.

**Immunocytochemistry.** Adult fly heads were placed in phosphate-buffered saline (PBS) containing 4% paraformaldehyde (PFA), incubated overnight at 4 °C, washed three times with PBS, transferred to 30% sucrose in PBS, and incubated overnight. The heads were frozen in Tissue-Tek Cryomold (Electron Microscopy Sciences, Cat. No. 62534-15), cut into 15-μm sections, and mounted on glass slides. Sections were permeabilized with 1% SDS for 5 min, washed three times with PBS, and placed in PBS with 10% goat serum and 10% donkey serum to block non-specific binding. After incubation in primary and secondary antibodies, sections were mounted in Vectashield Antifade Mounting Medium with DAPI (Vector Labs, Cat. No. H-1200). Images were acquired with a Leica TCS SP5 II laser-scanning confocal microscope.

Neurons were fixed in 4% PFA for 10 min and permeabilized with 0.2% Triton X-100 for 5 min. After incubation with 3% bovine serum albumin for 30 min, cells were incubated with primary antibodies overnight at 4 °C. After three washes with PBS, the cells were incubated with the secondary antibodies for 1 h at room temperature and coverslips were mounted in Vectashield Hardset Antifade Mounting Medium with DAPI (Vector Labs, Cat. No. H-1500). Images were acquired with a Zeiss LSM 800 microscope coupled to an AxioCam 506 camera (Zeiss). On average 300 cells were analyzed per experimental condition, $n = 3$ independent cultures.

The primary antibodies were rabbit-anti-poly(GR) (1:500; Millipore, Cat. No. MABN778), rat-anti-ELAV (1:50; Developmental Studies Hybridoma Bank, Cat. No. 7E8A10), chicken anti-GFP (1:500; Thermo Fisher Scientific, Cat. No. A10262), rabbit-anti-cleaved caspase-3 (1:250; Cell Signaling Technologies, Cat. No. 9661S), rabbit anti-*Drosophila* p62 (1:250, Abcam, Cat. No. ab178440), mouse anti-βIII-tubulin (1:200, Promega, Cat. No. G7121), and rabbit anti-TDP-43 (1:100, Protein Tech Group, Cat. No. 12892-1-AP), and mouse anti-MAP2 (1:500, Sigma, Cat. No. M9942). Rabbit polyclonal antibody against TBPH was generated with CQSSG peptide by Covance and used at 1:500. The secondary antibodies (all from Thermo Fisher Scientific) were goat anti-rat Alexa Fluor 546 (1:500; Cat. No. A-11081), goat anti-rat Alexa Fluor 488 (1:500; Cat. No. A-11006), donkey anti-rabbit Alexa Fluor 568 (1:500; Cat. No. A10042), and donkey anti-rabbit Alexa Fluor 488 (1:500; Cat. No. A21206).

**Western blotting.** Adult heads were lysed in RIPA buffer (Thermo Fisher Scientific, Cat. No. 89900) containing EDTA-free Halt Protease and Phosphatase Inhibitor Cocktail (100×) (Thermo Fisher Scientific, Cat. No. 78441), incubated on ice for 15 min, and centrifuged at 4 °C for 15 min at $16,000 \times g$ to obtain the soluble fraction. The pellet was washed three times with RIPA buffer, incubated for 15 min in 7 M urea buffer, and centrifuged at 4 °C for 15 min at $16,000 \times g$ to obtain the insoluble fraction. Proportional amounts of the insoluble and soluble fractions were loaded on the gel. Protein (10 μg) in the soluble fraction was run on a 10% SDS-PAGE gel and immunoblotted with rabbit anti-TBPH (1:1000), mouse anti-tubulin (1:200; Sigma, Cat. No. T6199), or rabbit anti-*Drosophila* p62, (1:2000, Abcam, Cat. No. ab178440) overnight at 4 °C. The immunoblots were then washed and

incubated with goat anti-rabbit IgG (H + L) (1:5000; Thermo Fisher Scientific, Cat. No. 31460) and goat anti-mouse IgG (H + L) (1:5000; Jackson ImmunoResearch, Cat. No. 115-035-003), both conjugated with horseradish peroxidase. All blocking and antibody dilutions were done in 5% milk. Signals were developed with ECL Plus (Thermo Fisher Scientific, Cat. No. 32132). Uncropped and unprocessed scans of the blots can be found in the Source Data file.

**RNA fluorescence in situ hybridization.** Third instar fly larval brains were fixed in 4% PFA for 20 min, washed three times for 5 min each with PBS, permeabilized with 0.5% Triton X-100, and incubated in prehybridization buffer consisting of 2× saline sodium citrate (SSC), 40% formamide, salmon sperm DNA (1 mg/ml), and 0.1% Tween-20 for 1 h at 56 °C. Samples were then incubated in hybridization buffer containing 5′-end Cy3-conjugated $(G_2C_4)_4$ oligonucleotide probe for 3 h at 56 °C. Samples were then washed twice in wash buffer 1 (40% formamide, 2× SSC, and 0.1% Tween-20) for 15 min at 56 °C and twice more in wash buffer 2 (2× SSC and 0.1% Tween-20) for 15 min at room temperature, rinsed with PBS with 0.1% Tween-20, and mounted in Vectashield Antifade Mounting Medium with DAPI (Vector Labs).

iPSCs and neurons on glass coverslips were fixed in 4% PFA for 20 min, permeabilized in 70% ethanol overnight at 4 °C, incubated with 40% formamide/2× SSC for 10 min at room temperature, and hybridized for 2 h at 56 °C with the Cy3-conjugated probe in hybridization buffer consisting of 40% formamide, 2× SSC, 10% dextran sulfate, yeast tRNA (1 mg/ml), and salmon sperm DNA (1 mg/ml). The cells were washed once with 40% formamide/1× SSC for 30 min at 37 °C and twice with 1× SSC at room temperature for 30 min. Neurons were further incubated with 3% bovine serum albumin for 30 min and then immunostained for the neuronal marker MAP2 following the immunocytochemistry protocol as described above. Coverslips were mounted in Vectashield Hardset Antifade Mounting Medium with DAPI (Vector Labs, Cat. No. H-1500). Images were acquired with a Zeiss LSM 800 microscope and an AxioCam 506 camera (Zeiss). On average 100 cells were analyzed per experimental condition, $n = 3$ independent cultures.

**Measurement of poly(GR) and poly(GP).** Poly(GR) and poly(GP) were measured with a MSD immunoassay. Briefly, 2-week-old fly heads were homogenized with a pellet pestle motor in Tris lysis buffer (MSD; Cat. No. R60TX-3) containing protease and phosphatase inhibitor cocktail (Thermo Fisher Scientific, Cat. No. 78441), sonicated on ice at a 20% pulse rate for 20 s, and centrifuged at $16,000 \times g$ for 15 min at 4 °C. The protein concentration of the lysates was determined with the BCA protein assay (Thermo Fisher Scientific; Cat. No. 23225). iPSCs and neurons were lysed in ice-cold RIPA buffer (Thermo Fisher Scientific; Cat. No. 89900) containing a cocktail of protease and phosphatase inhibitors (Thermo Fisher Scientific; Cat. No. 78441), sonicated on ice at a 20% pulse rate for 15 s, and centrifuged at $16,000 \times g$ for 20 min at 4 °C. The protein content of supernatants was determined with the Bio-Rad Protein assay reagent (Bio-Rad; Cat. No. 500–0006). Fly samples (0.9 μg/μl) and iPSCs and neurons (2 μg/μl) samples were loaded on a 96-well single-spot plate (MSD; Cat. No. L45XA) precoated with custom-made polyclonal rabbit anti-$(GR)_8$ or anti-$(GP)_8$ antibodies (1 μg/ml, Covance) and tested in duplicate wells. Serial dilutions of recombinant $(GR)_8$ peptide in 1% BSA-TBST were used to prepare the standard curve. The detection antibodies were anti-$(GR)_8$ or anti-$(GP)_8$ antibodies previously tagged with GOLD SULFO (GOLD SULFO-TAG NHS-Ester Conjugation Pack, MSD; Cat. No. R31AA) at a concentration of 0.5 μg/ml. Response signals from the assay plate were acquired with a QuickPlex SQ120 instrument (MSD). For background correction, values from control flies (*CONT-(GR)80*) or values from iPSCs/neurons from a control subject were subtracted from the corresponding test samples. For parental and respective knockout lines from patient 1, the poly(GP) levels were initially measured with an alternative anti-$(GP)_8$ antibody as described before[57].

**Axonal degeneration index.** Axonal degeneration was analyzed by immunostaining for βIII-tubulin (TUJ1)[58]. Briefly, images of 6–8 random fields were acquired with a 20× objective lens using a Zeiss LSM 800 microscope coupled to an AxioCam 506 camera and analyzed with the ImageJ software. To detect degenerated axons, we used the particle analyzer module of ImageJ on binarized images to calculate the area of the small fragments or particles (size 20–10,000 pixels). The axonal degeneration index was defined as the ratio of the area of degenerated axons to the total axon area (healthy plus degenerated axons). In the graph, each independent data point represents one field, 6–8 fields were collected per culture/differentiation, $n = 3$ independent cultures.

**Transmission EM.** Three-week-old fly brains were dissected in PBS and immediately immersion fixed in 2.5% glutaraldehyde in 0.1 M sodium cacodylate buffer, pH 7.2. Samples were processed and analyzed by standard procedures at the Electron Microscopy Facility at the University of Massachusetts Medical School. Briefly, fixed samples were incubated overnight in fresh 2.5% glutaraldehyde in 0.1 M sodium cacodylate buffer at 4 °C, rinsed twice in the same fixation buffer, and postfixed with 1% osmium tetroxide for 1 h at room temperature. Samples were then washed twice for 5 min each with distilled water and dehydrated through a graded ethanol series (20% increments), before two changes in 100% ethanol.

Samples were infiltrated first with two changes of 100% propylene oxide and then with a 50:50 mixture of propylene oxide and SPI-Pon 812 resin. The next day, after five changes of fresh 100% SPI-Pon 812 resin, the samples were polymerized at 68 °C in flat embedding molds. The samples were reoriented, cut into ~70-nm sections ~100 μm from the tip of the ventral ganglion or central brain (frontal sections), placed on copper support grids, stained with lead citrate and uranyl acetate, and examined with a FEI Tecnai 12 BT electron microscope at 100 Kv accelerating voltage. Images were captured with a Gatan TEM CCD camera.

**Reporting summary**. Further information on research design is available in the Nature Research Reporting Summary linked to this article.

## Data availability

The data that supports the findings of this study are available from the corresponding authors upon request. Source data underlying Figs. 1–9 and Supplementary Figs. 1, 3–11, and 14–19 are available as a Source Data file. All unique materials such iPSC lines are readily available upon execution of a Material Transfer Agreement.

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

## Acknowledgements

We thank Dr E. Hafen, the Bloomington Drosophila Stock Center, and the VDRC for fly lines. We also thank Dr O. King for help with statistical analysis. This work was supported by the National Institutes of Health (R01NS101986, R01NS093097, R37NS057553, and R21NS109847 to F.-B.G.), and grants from the Packard Center for ALS Research, the Target ALS Foundation, and the Muscular Dystrophy Association (F.-B.G.), and the Frick Foundation for ALS Research, the ALS Association, the Angel Fund, and the Alzheimer's Association (2016-NIRG-396129) (S.A.). The UMMS EM Core was supported by grant S10RR027897 from the National Center for Research Resources. The content is solely the responsibility of the authors and does not necessarily represent the official views of the National Center for Research Resources or the National Institutes of Health.

## Author contributions

Y.Y.-A. performed all the *Drosophila* experiments and foci analysis in *C9ORF72* iPSCs, S.A. did all other experiments on iPSCs and iPSC-derived neurons. S.A., G.K. and T.F.G. did DPR ELISA analyses. Y.Y.-A., S.A. and F.-B.G. analyzed the data and wrote the manuscript. F.-B.G. supervised the project.

## Competing interests

The authors declare no competing interests.
