## [Peer Review File · Nature Communications]

Reviewers' comments:

Reviewer #1 (Remarks to the Author):

The manuscript by Yeliz and colleagues performed an unbiased genetic screen that led to the identification of Lilli as a strong modifier of poly(GR) toxicity in vivo. They observed that knocking out AFF2/FMR2 (human homologue of Lilli) decreased the expression of the mutant C9ORF72 allele containing expanded expanded G4C2 repeats, levels of RNA foci as well as dipeptide proteins in cortical neurons from C9 iPSC lines. Overall, this is an important study that shows identification of Lilli as a dominant modifier of C9 toxicity. Here are few concerns that the authors should consider addressing.

1. The authors claim that poly(GR) expression is increased in an age-dependent manner (Fig 1B). This is an overstatement as they did not perform any quantification. They should perform WB or dot blot along with quantification to support their claim.
2. They stated that "Despite the axonal degeneration, only 0.7% of (GR)⁸⁰-expressing neurons in 21-day...". It is not clear where this 0.7% number is coming from? There is no quantification data here and should be included.
3. In figure 2d, they showed that co-expression of the viral antiapoptotic gene p35 did not extend the life span. There is no control here. They should include the life span of p35 overexpression itself.
4. The authors claim that they did not observe any TDP-43 pathology in poly(GR) expression model and suggest that neurodegenerative phenotypes appear without any TDP-3 pathology. This is contradictory to human ALS/FTD patient data where TDP-43 pathology is reported in 97% of ALS patients and over 50% dementia patients. The authors should provide some alternative explanations for this. What is the application of this artificial model that does not recapitulate a key pathological feature fully accepted in ALS/FTD field. Is it possible that the TDP-43 pathology originate by mixture of all 5 DPRs and the authors do not see any TDP-43 pathology because they just expressed only poly(GR)?
5. The authors should provide data showing loss of Lilli activity to support their claim (line 142).
6. Is Lilli specific for poly(GR) toxicity? Does it suppress other ALS/FTD-causing genes?
7. The authors should include IF panel for Lilli mutant allele in figure 5d-e. It is not clear why this data has been not included.
8. What happens to TDP-43 pathology in C9 iPSC lines with and without AFF2/FMR2 KO? This is the most obvious experiment that the authors should have done to support their claim.
9. Figure S4: The authors have not properly characterized the anti-TBPH antibody that they generated. It is not clear why tubulin levels are reduced in different lines/samples? The authors should use TBPH null or RNAi to demonstrate the specificity of their antibody.

Reviewer #2 (Remarks to the Author):

Summary:

Yuva-Aydemir et al. identified AFF2/FMR2, a member of the super elongation complex, as a modifier of C9ORF72 repeat toxicity in *Drosophila*. Their data suggest that AFF2 modifies C9 repeat toxicity by downregulating the repeat RNA. They then demonstrate that downregulating AFF2 in iPSCs from C9-ALS patients decreases the C9-V3 transcript containing GGGGCC repeats. Because repeat expression is toxic, lowering the number of repeat-containing transcripts would presumably mitigate neurodegeneration; however, downregulating GGGGCC transcripts by targeting AFF2 may not be an optimal therapeutic target since mutations in AFF2 are linked to fragile XE syndrome. The authors' manuscript would certainly be of interest to the ALS field, especially since it complements previous work identifying AFF2 as a modifier of TDP43 toxicity (Chung et al., 2018). However, there are several concerns regarding quantification and statistical analyses that need to be addressed.

Major Comments:

- The variance of mRNA quantity for control samples is not shown in the majority of figures.
- Double normalization (eg. for qPCR, normalizing first to a reference gene and then normalizing that to a control genotype/timepoint) can mask important aspects of the data. It would be better to only normalize a single time for relativistic measurements (eg. report the ratio of “gene of interest” to “reference gene” without also normalizing to a control genotype).
- ANOVA generally assumes variance is equal across groups, but this does not appear to be true given the way mRNA quantification is currently shown. Please ensure that your data meets the assumptions of the statistical tests being used.
- The authors use ANOVA tests several times, but the figures show comparisons between individual genotypes. Is a post-hoc test being used? Which one?
- The authors suggest that GR dipeptide repeats cause neurite degeneration without substantial neuronal death, but the evidence for neurodegeneration in Figure 2 is not very strong. Driving membrane-bound GFP and examining leg motor neurons may be a useful complementary approach, but may not be necessary if the current data is more rigorously quantified. This is particularly a concern with the axonal degeneration EM images. The assertion in the discussion linking C9 to axonal degeneration is not sufficient without better quantification and would be strongly supported by a similar finding in the iPSC neurons.

Specific Comments:

- Figure 1:
 - o The diagram in Fig. 1a is helpful, but the control construct is not described in the text.
 - o What is the exact sequence of the UAS constructs? This can be referenced or listed in the supplement.
 - o Fig. 1b- Quantification of poly(GR) is missing.
 - o Fig. 1c- The variance of control mRNA levels (week1) needs to be shown. This is a problem throughout the manuscript.
- Figure 2:
 - o The labeling in Fig. 2a and 2b is confusing since Fig. 2a has subpanels also labeled as “A” and “B.” In the figure legend these are lowercase “a” and “b.” Changing the naming convention to something such as “Fig. 2a, i” would be helpful.
 - o Quantification of MLBs in Fig. S2 should be moved to the main figure. Is this quantification from the brain or VNC? How many animals were sampled?
 - o How were the MLBs quantified? If they were quantified manually, were images sampled and quantified in a blinded fashion?
 - o Quantification of neurodegeneration is needed.
 - o Quantification for CC3-positive neurons is described, but not shown. How many cells and animals were sampled?
 - o How was the quantification of insoluble dTDP43 normalized? Please add this information to the methods.
 - o In Fig. 2g, there is a large variance in dTDP43 protein levels which prohibits detecting changes between genotypes with such small sample numbers. Even so, there appears to be a trend toward decreased soluble dTDP43 even with n=3. Please consider a power analysis based on the observed variances, for example ensuring 80% power to detect a 2 SD difference requires n=5 /group.
- Figure 3:
 - o It is unclear what the mutation is for allele A17-2. What is the mutation and how do we know it is a loss of function?
 - o Figure S6 was helpful for describing the deletion alleles used. Please add the Lilli alleles to this figure.
 - o Please add more information about the Lilli alleles to the text.
- Figure 4:
 - o The use of 3 different Lilli alleles across figures is rigorous and convincing, but Lilli[4US] is missing in panel A. Please add it.
 - o Fig. 4a- GR(80)/w1118 mRNA variance needs to be shown. mRNA variance for controls is shown in panels d and e.
- Figure 5:

- o W1118 is currently written as W1111[]
- o Please add to the text that motor neurons are being measured in Fig 5d.
- o In Fig 5d, please label the green channel as GFP
- o How many animals were used for the quantification in Fig. 5e? Are comparisons still significant if measurements are pooled by animal?
- Figures 6:
 - o Similar problems with displaying qPCR control data and only normalizing a single time would be preferred.
 - o Please clarify the sex of the iPSCs in the text given the location of AFF2.
- Figure 7:
 - o Similar problems with displaying qPCR control data and only normalizing a single time would be preferred.
 - o Why is figure 7g normalized to 0.5 while everything else is normalized to 1?
 - o The authors do not explain why C9 antisense is being measured, or whether the measurement is to antisense C9 repeats or antisense to the C9ORF72 gene.
- Figure 8:
 - o Representative images need to be shown. There are images in Fig. S14, but they are not representative of the quantification.
- Figure S1:
 - o The control lifespans in Fig S1A and B are quite different. If both the experimental and control genotypes for each figure were run at the same time, then this is not a major issue. If they were not run at the same time though, then the dramatic difference in control lifespans suggests that variance due to technical differences could be a major factor.
- Figure S3:
 - o Immunofluorescence images of p62 and western blots of p62 are shown, but the antibody used is not described in the methods.
- Figure S11:
 - o Data for AFF4 is described in the text but not shown in Fig S11.
 - o How many other reference genes were tested? The text mentions “all the reference genes tested,” but there is no indication as to how many or which were tested.
- Text edits:
 - o Line 64: This is not so much a new “function” of AFF2; AFF2 was previously known to regulate transcription, but not the C9ORF72 transcript with GGGGCC repeats.
 - o Line 91: The control construct is not described in the text.
 - o Line 188: For clarity, please state in the text that foci measurements are in motor neurons.
 - o Citation #35 can be updated

Discussion points:

- It is interesting that Lilli was also reported in a TDP43 screen (Chung et al., 2018), but apparently was not a hit from several C9 screens (Freibaum et al., 2015; Jovičić et al., 2015; Zhang et al., 2015; Boeynaems et al., 2016; Lee et al., 2016). It would be interesting to 1) discuss whether your screen found genes that have been previously identified as C9 modifiers and 2) discuss why your screen may be identifying hits not seen in other C9 screens.

Reviewer’s References

- Boeynaems S et al. (2016) Drosophila screen connects nuclear transport genes to DPR pathology in c9ALS/FTD. *Sci Rep* 6:20877.
- Chung CY, Berson A, Kennerdell JR, Sartoris A, Unger T, Porta S, Kim HJ, Smith ER, Shilatifard A, Van Deerlin V, Lee VMY, Chen-Plotkin A, Bonini NM (2018) Aberrant activation of non-coding RNA targets of transcriptional elongation complexes contributes to TDP-43 toxicity. *Nat Commun* 9.
- Freibaum BD, Lu Y, Lopez-Gonzalez R, Kim NC, Almeida S, Lee K-H, Badders N, Valentine M, Miller BL, Wong PC, Petrucelli L, Kim HJ, Gao F-B, Taylor JP (2015) GGGGCC repeat expansion in C9orf72 compromises nucleocytoplasmic transport. *Nature* 525:129–133.
- Jovičić A, Mertens J, Boeynaems S, Bogaert E, Chai N, Yamada SB, Paul JW, Sun S, Herdy JR, Bieri G, Kramer NJ, Gage FH, Van Den Bosch L, Robberecht W, Gitler AD (2015) Modifiers of C9orf72

dipeptide repeat toxicity connect nucleocytoplasmic transport defects to FTD/ALS. *Nat Neurosci* 18: 1226–1229.

Lee KH et al. (2016) C9orf72 Dipeptide Repeats Impair the Assembly, Dynamics, and Function of Membrane-Less Organelles. *Cell* 167: 774-788.e17.

Zhang K et al. (2015) The C9orf72 repeat expansion disrupts nucleocytoplasmic transport. *Nature* 525: 56–61.

Reviewer 1:

1. *The manuscript by Yeliz and colleagues performed an unbiased genetic screen that led to the identification of Lilli as a strong modifier of poly(GR) toxicity in vivo. They observed that knocking out AFF2/FMR2 (human homologue of Lilli) decreased the expression of the mutant C9ORF72 allele containing expanded G4C2 repeats, levels of RNA foci as well as dipeptide proteins in cortical neurons from C9 iPSC lines. Overall, this is an important study that shows identification of Lilli as a dominant modifier of C9 toxicity.*

We thank the reviewer for his/her positive comments and constructive comments below.

2. *The authors claim that poly(GR) expression is increased in an age-dependent manner (Fig 1B). This is an overstatement as they did not perform any quantification. They should perform WB or dot blot along with quantification to support their claim.*

The conclusion that poly(GR) expression is increased in an age-dependent manner was initially based on the observation from the immunostaining results that showed an age-dependent increase in the number of neurons expressing detectable poly(GR) (Fig. 1B). More recently, we established a sensitive ELISA assay (Choi et al., *Nat. Neurosci.* 2019), which we used to quantify poly(GR) levels as the reviewer suggested. This new result is now presented in Fig. 1c.

3. *They stated that “Despite the axonal degeneration, only 0.7% of (GR)80-expressing neurons in 21-day...”. It is not clear where this 0.7% number is coming from? There is no quantification data here and should be included.*

As the reviewer suggests, the quantification data is now included in Fig. 2e.

4. *In figure 2d, they showed that co-expression of the viral antiapoptotic gene p35 did not extend the life span. There is no control here. They should include the life span of p35 overexpression itself.*

It was reported by others that p35 expression itself did not affect the life span of flies. In the revised manuscript, we repeated this experiment and included the data in Fig. 2h.

5. *The authors claim that they did not observe any TDP-43 pathology in poly(GR) expression model and suggest that neurodegenerative phenotypes appear without any TDP-3 pathology. This is contradictory to human ALS/FTD patient data where TDP-43 pathology is reported in 97% of ALS patients and over 50% dementia patients. The authors should provide some alternative explanations for this. What is the application of this artificial model that does not recapitulate a key pathological feature fully accepted in ALS/FTD field. Is it possible that the TDP-43 pathology originate by mixture of all 5 DPRs and the authors do not see any TDP-43 pathology because they just expressed only poly(GR)?*

The lack of TDP-43 pathology in poly(GR)-expressing flies is consistent with similar observations in poly(GR)-expressing mice from the Petrucelli lab and our lab (Zhang et al., *Nat. Med.* 2018; Choi et al., *Nat. Neurosci.* 2019). Rather than contradicting data from ALS/FTD patients, these findings in flies and mice suggest that poly(GR) alone is not sufficient to induce TDP-43 pathology. As the reviewer correctly points out, a combination of different DPR proteins may be needed to induce TDP-43 pathology; this possibility remains to be tested experimentally in the future. The reductionist approach of studying the toxicity of individual DPR proteins in flies and mice offers detailed mechanistic insights that help us understand the disease in a comprehensive way. As suggested by the reviewer, we have revised our Discussion section accordingly.

6. *The authors should provide data showing loss of Lilli activity to support their claim (line 142).*

We did a couple of experiments to confirm that the three *lilli* alleles are indeed loss-of-function alleles, as they reduced the expression of *lilli* mRNA (Supplementary Fig. 6b) or its target gene *Hsp70* (Supplementary Fig. 6c).

7. *Is Lilli specific for poly(GR) toxicity? Does it suppress other ALS/FTD-causing genes?*

Lilli suppresses TDP-43 toxicity without affecting its expression level (Chung et al., 2018), suggesting a mechanism different from its role in transcription elongation of expanded G₄C₂ repeats. We commented on this paper in the Discussion. As suggested by the reviewer, we did a new experiment, which showed that Lilli does not suppress the toxicity of FTD3-associated mutant CHMP2B, suggesting a certain degree of specificity. This new result is presented in Supplementary Fig 7.

8. *The authors should include IF panel for Lilli mutant allele in figure 5d-e. It is not clear why this data has been not included.*

We corrected this oversight.

9. *What happens to TDP-43 pathology in C9 iPSC lines with and without AFF2/FMR2 KO? This is the most obvious experiment that the authors should have done to support their claim.*

As suggested by the reviewer, we analyzed TDP-43 pathology in C9 iPSC-derived cortical neurons with or without *AFF2* KO. Cytoplasmic TDP-43 was found in similar numbers of young C9 neurons, C9 neurons in which the expanded G₄C₂ repeats were deleted by CRISPR-Cas9, and C9 neurons with *AFF2* KO (data presented in new Supplementary Fig. S18). This result was expected, as these neurons do not show obvious disease phenotypes until they have been in culture for several months. Because of time restrictions, we decided to repeat this experiment with neurons kept in the absence of neurotrophic factors (BDNF and GDNF). After 2–3 weeks without these factors, the number of C9 neurons with cytoplasmic TDP43 was increased, and the increase was largely prevented by *AFF2* KO (data presented in new Fig. 9).

10. *Figure S4: The authors have not properly characterized the anti-TBPH antibody that they generated. It is not clear why tubulin levels are reduced in different lines/samples? The authors should use TBPH null or RNAi to demonstrate the specificity of their antibody.*

The protein lysates for this western blot analysis came from 4 independent genetic crosses. Different amounts of total proteins were loaded on the gel depending on their availability. What is important is we loaded same amount of proteins from wildtype and *TBPH* KO flies. In each case, the newly generated TBPH antibody recognized the TBPH protein in wildtype but not *TBPH* KO flies. The *Tbph*^{Q367X} allele was generated in our lab and shown to be a null allele (Lu et al., 2009).

Reviewer 2:

1. *Yuva-Aydemir et al. identified AFF2/FMR2, a member of the super elongation complex, as a modifier of C9ORF72 repeat toxicity in Drosophila. Their data suggest that AFF2 modifies C9 repeat toxicity by downregulating the repeat RNA. They then demonstrate that downregulating AFF2 in iPSCs from C9-ALS patients decreases the C9-V3 transcript containing GGGGCC repeats. Because repeat expression is toxic, lowering the number of repeat-containing transcripts would presumably mitigate neurodegeneration; however, downregulating GGGGCC transcripts by targeting AFF2 may not be an optimal therapeutic target since mutations in AFF2 are linked to fragile XE syndrome. The authors' manuscript would certainly be of interest to the ALS field, especially since it complements previous work identifying AFF2 as a modifier of TDP43 toxicity (Chung et al., 2018).*

We thank the reviewer for his/her appreciation of our work and exceptionally detailed constructive comments below.

2. *The variance of mRNA quantity for control samples is not shown in the majority of figures. Double normalization (eg. for qPCR, normalizing first to a reference gene and then normalizing that to a control genotype/timepoint) can mask important aspects of the data. It would be better to only normalize a single time for relativistic measurements (eg. report the ratio of "gene of interest" to "reference gene" without also normalizing to a control genotype).*

We revised most of the figures to show the variance of mRNA levels for control samples. We think it is better not to show this for the experiments in Fig. 5b,c, as genetic crosses and mRNA measurement were done multiple times over a period of several months. Please see our response to comment 26 for a more detailed explanation.

3. *ANOVA generally assumes variance is equal across groups, but this does not appear to be true given the way mRNA quantification is currently shown. Please ensure that your data meets the assumptions of the statistical tests being used.*

The reviewer is correct that ANOVA generally assumes equal variance across groups. As stated in our response to Comment 2, we revised most of the figures to show the variance

of mRNA levels for control samples and continue to use ANOVA as the statistical test. For some other panels that show normalization to a control genotype, we used a *t* test without assuming that variance was equal across groups.

4. *The authors use ANOVA tests several times, but the figures show comparisons between individual genotypes. Is a post-hoc test being used? Which one?*

We used Dunnett's test as a post-hoc test, as now stated in figure legends.

5. *The authors suggest that GR dipeptide repeats cause neurite degeneration without substantial neuronal death, but the evidence for neurodegeneration in Figure 2 is not very strong. Driving membrane-bound GFP and examining leg motor neurons may be a useful complementary approach, but may not be necessary if the current data is more rigorously quantified. This is particularly a concern with the axonal degeneration EM images. The assertion in the discussion linking C9 to axonal degeneration is not sufficient without better quantification and would be strongly supported by a similar finding in the iPSC neurons.*

As suggested by the reviewer, we carefully quantified neurodegeneration in Fig. 2; the results are shown in a new panel (Fig. 2g). More importantly, as suggested by the reviewer, we analyzed axonal degeneration in C9 iPSC-derived cortical neurons vs. C9 neurons in which the expanded G₄C₂ repeats were deleted with CRISPR-Cas9 technology. In the absence of neurotrophic factors, C9 neurons had increased axonal degeneration, a phenotype that was largely prevented by *AFF2* KO (data presented in new Fig. 9).

6. *The diagram in Fig. 1a is helpful, but the control construct is not described in the text. What is the exact sequence of the UAS constructs? This can be referenced or listed in the supplement.*

In Page 4 of the revised manuscript, we describe the control construct and cite our earlier work (Yang et al., *Acta Neuropathol.* 2015) in which these UAS constructs were first published.

7. *Fig. 1b- Quantification of poly(GR) is missing.*

We recently established a sensitive ELISA (Choi et al., *Nat. Neurosci.* 2019), so we could quantify poly(GR) levels as the reviewer suggests. This new result is presented in Fig. 1c.

8. *Fig. 1c- The variance of control mRNA levels (week1) needs to be shown. This is a problem throughout the manuscript.*

The variance of control mRNA levels is now shown in revised Fig. 1. In the experiment in Fig. 1d (originally Fig. 1c), we sought to compare the relative (*GR*)₈₀ mRNA levels in flies of different ages from the same genetic cross, measured side by side under identical quantitative PCR conditions. Then the same genetic cross and mRNA measurement were repeated independently 6 times. The variance of control mRNA levels could reflect

technical issues, since genetic crosses and mRNA measurement were done at different times over a period of several months.

In addition, we further clarify in the legend how this experiment was done. More importantly, most of the figures throughout the manuscript have been revised to show the variance of control levels wherever appropriate.

9. *The labeling in Fig. 2a and 2b is confusing since Fig. 2a has subpanels also labeled as “A” and “B.” In the figure legend these are lowercase “a” and “b.” Changing the naming convention to something such as “Fig. 2a, i” would be helpful.*

Thank you for the helpful suggestion. We have revised Fig. 2 accordingly.

10. *Quantification of MLBs in Fig. S2 should be moved to the main figure. Is this quantification from the brain or VNC? How many animals were sampled?*

We moved EM analysis and quantification of MLBs to Fig. 2. The quantification was from the brain, and 3 flies per genotype were quantified.

11. *How were the MLBs quantified? If they were quantified manually, were images sampled and quantified in a blinded fashion?*

Yes, it was done in a blinded fashion. We state so clearly in the legend section.

12. *Quantification of neurodegeneration is needed.*

In the revised manuscript, the quantification of degenerated axons is presented in new Figure 2e.

13. *Quantification for CC3-positive neurons is described, but not shown. How many cells and animals were sampled?*

Quantification for CC3-positive neurons is now presented in new Fig. 2g. We state in the legend that “Six brains per genotype were quantified. Dots represent the average percentage of cleaved caspase 3–positive neurons out of 200–250 neurons.”

14. *How was the quantification of insoluble dTDP43 normalized? Please add this information to the methods.*

As presented below in our response to comment 15, we repeated the western blot analysis, and found that much more consistent levels of insoluble dTDP43 in independent experiments. We state in the Methods that proportional amounts of the insoluble and soluble fractions were loaded on the gel.

15. *In Fig. 2g, there is a large variance in dTDP43 protein levels which prohibits detecting changes between genotypes with such small sample numbers. Even so, there appears to*

be a trend toward decreased soluble dTDP43 even with n=3. Please consider a power analysis based on the observed variances, for example ensuring 80% power to detect a 2 SD difference requires n=5 /group.

We apologize for the poor quality of this experiment. We repeated western blot analysis and found that the relative dTDP43 protein levels are much more consistent between independent experiments. This new result indicates that indeed (GR)₈₀ does not affect the levels of soluble or insoluble dTDP43 (new Fig. 2j, k).

16. It is unclear what the mutation is for allele A17-2. What is the mutation and how do we know it is a loss of function?

The nature of the mutation in A17-2 is unclear in the literature. To confirm it is a loss of function allele, we measured its effect on the expression level of the *Lilli* target gene *Hsp70* (Lin et al., *Mol. Cell* 2010). Like the two other *lilli* alleles, A17-2 also decreased *Hsp70* level, indicating it is indeed a loss of function allele. This new result is now presented in Supplementary Fig. 6c.

17. Figure S6 was helpful for describing the deletion alleles used. Please add the Lilli alleles to this figure.

We added two *Lilli* alleles (except A17-2) to Supplementary Fig. 6a.

18. Please add more information about the Lilli alleles to the text.

We added more information about the *Lilli* alleles to the Methods section.

19. The use of 3 different Lilli alleles across figures is rigorous and convincing, but Lilli[4US] is missing in panel A. Please add it.

We added *Lilli[4U5]* to Panel a.

20. Fig. 4a- GR(80)/w1118 mRNA variance needs to be shown. mRNA variance for controls is shown in panels d and e.

In revised manuscript, *GR(80)/w1118* mRNA variance is shown in Fig. 4a.

21. W1118 is currently written as W111[]

The error has been corrected.

22. Please add to the text that motor neurons are being measured in Fig 5d.

Done.

23. In Fig 5d, please label the green channel as GFP.

Done.

24. *How many animals were used for the quantification in Fig. 5e? Are comparisons still significant if measurements are pooled by animal?*

We now state in the Fig. 5e legend that “5-7 motor neurons from seven third instar larvae per genotype were quantified.” Yes, measurements are pooled by animal and comparisons are still significant.

25. *Similar problems with displaying qPCR control data and only normalizing a single time would be preferred.*

In this panel, each dot is an independent genetic cross. For each biological replicate, crosses of three different genotypes were set up at the same time, and qPCR measurement for these samples was done side by side under the same condition. Then this experiment was repeated multiple times over several months. Thus, we think it is better to normalize against wildtype in each biological replicate. In this case, we agree with the reviewer that ANOVA is not the best statistical test. Instead, we use Welch’s *t* test in the revised Fig. 5b, c.

26. *Please clarify the sex of the iPSCs in the text given the location of AFF2.*

The first parental iPSC line is from a female donor and the second is from a male donor. Given the location of *AFF2* on the X chromosome, we only used homozygous *AFF2* deletion lines from the female donor to avoid issues associated with X-inactivation. This important point has been clarified in the revised manuscript.

27. *Similar problems with displaying qPCR control data and only normalizing a single time would be preferred.*

For this experiment, each set of samples came from one independent neuronal differentiation of iPSC lines and the qPCR measurements were done at the end of each differentiation. Since each differentiation was set to be 4-6 weeks apart, different qPCR measurements were also 4-6 weeks apart. For this reason, we believe it is better to normalize to the respective parental control in each differentiation to take into account possible variations introduced by using reagents from different lots and, in some cases, different providers. Nonetheless, we also looked into the possibility of re-running the samples from all sets of differentiations in the same qPCR plate. Unfortunately, we do not have enough materials to do so.

28. *Why is figure 7g normalized to 0.5 while everything else is normalized to 1?*

The error has been corrected.

29. *The authors do not explain why C9 antisense is being measured, or whether the measurement is to antisense C9 repeats or antisense to the C9ORF72 gene.*

Our goal was to test the effect of *AFF2* knock-down on the transcription of *C9ORF72* RNAs. The first intron of the *C9ORF72* gene can be transcribed in the antisense direction (Zu et al., *PNAS*, 2013) and in the sense direction. Therefore, we used qPCR assays to measure both the sense and antisense RNAs transcribed from both intron 1, which contains repeats, and variant 2, which does not. This point is clarified in the current version of the manuscript.

30. *Figure 8: Representative images need to be shown. There are images in Fig. S14, but they are not representative of the quantification.*

Representative images are now presented in Figure 8 and Supplementary Fig. 15 (old Fig. S14).

31. *The control lifespans in Fig S1A and B are quite different. If both the experimental and control genotypes for each figure were run at the same time, then this is not a major issue. If they were not run at the same time though, then the dramatic difference in control lifespans suggests that variance due to technical differences could be a major factor.*

The flies in Fig. S1a and S1b had different genetic backgrounds and were raised at different temperatures. Fig. 1a is for flies with *Elav-Gal4* and grown at 29° C, while Fig. 1b is for flies with *OK-371-Gal4* and grown at 25° C. Therefore, these control flies have a significant difference in their lifespans. To make this point clear, both genotype and temperature are indicated on the top of Fig. S1a and S1b.

32. *Immunofluorescence images of p62 and western blots of p62 are shown, but the antibody used is not described in the methods.*

The antibody is now specified in the Methods section.

33. *Data for AFF4 is described in the text but not shown in Fig S11.*

The error has been corrected. Data for AFF4 is described in Fig. S12 (old Fig. S11).

34. *How many other reference genes were tested? The text mentions “all the reference genes tested,” but there is no indication as to how many or which were tested.*

The reference genes tested were *ACTN*, *GAPDH*, *HPRT* and *SUPT5H* (as suggested by the Genevestigator software). This information was originally included in the legend of the Supplementary Figure 11 and now has been added to the respective Materials and Methods section of the revised manuscript.

35. *Line 64: This is not so much a new “function” of AFF2; AFF2 was previously known to regulate transcription, but not the C9ORF72 transcript with GGGGCC repeats.*

We rewrote the sentence as follows: “Thus, AFF2/FMR2 helps regulate the transcripts of expanded G4C2 repeats in human *C9ORF72*-ALS/FTD neurons.”

36. *Line 91: The control construct is not described in the text.*

The control construct is now described in the revised text.

37. *Line 188: For clarity, please state in the text that foci measurements are in motor neurons.*

These are iPSC-derived cortical neurons. The text has been revised accordingly.

38. *Citation #35 can be updated.*

Done.

39. *It is interesting that Lilli was also reported in a TDP43 screen (Chung et al., 2018), but apparently was not a hit from several C9 screens (Freibaum et al., 2015; Jovičić et al., 2015; Zhang et al., 2015; Boeynaems et al., 2016; Lee et al., 2016). It would be interesting to 1) discuss whether your screen found genes that have been previously identified as C9 modifiers and 2) discuss why your screen may be identifying hits not seen in other C9 screens.*

We already cited work by Chung et al. (*Nat. Commun.* 2018) in which they found that Lilli modifies TDP-43 toxicity. We also added a recently published study by Goodman et al. (*Nat. Neurosci.* 2019) that shows that the PAF1 complex, a positive regulator of RNA polymerase II transcription elongation, modifies G₄C₂ repeat toxicity by regulating the transcription of expanded G₄C₂ repeats in *Drosophila*.

In a deficiency screen, we identified several deficiency lines as modifiers of poly(GR) toxicity. We quickly focused on Lilli because the number of genes in the overlapping region of multiple deficiencies is rather small, which allowed us identify Lilli as the first suppressor gene from this screen. We have not identified the genes responsible for the modifier effects of other deficiency lines. Thus, at this moment, we do not know whether our screen will discover genes previously identified as C9 modifiers. Our screen may identify genes not seen in other screens, as we expressed poly(GR) in adult neurons, possibly avoiding the toxicity of poly(GR) during development. Moreover, we used lifespan as the readout, which may lead to the identification of genes not seen in other genetic screens that use the rough eye phenotype as the readout.

As suggested by the reviewer, we added a couple of sentences in the Discussion to speculate that novel modifiers not seen in other genetic screens may be identified from other deficiency lines identified in our study.

Reviewers' comments:

Reviewer #1 (Remarks to the Author):

The authors have addressed the concerns raised by the reviewers satisfactorily. The revised manuscript is significantly improved. I recommend this manuscript for the publication without any reservation.

Reviewer #2 (Remarks to the Author):

The authors have done a great job of addressing most concerns, but there are still problems with the statistical analyses and a couple concerns regarding new data presented in Figure 9.

Major Revisions:

- There are still major concerns about the statistical approach and we recommend that the authors consult with a statistician. Double normalization as performed by the authors in multiple key figures results in 0 variance in the control group, breaking the assumption of normality required for a Welch's t-test. We might perform a paired t-test based on the authors' description of potentially different experimental conditions, but we strongly recommend that the authors consult a statistician to determine which statistical approaches are appropriate. This is particularly a concern when a small sample size ($n=3$ per group frequently) presumably yields a poor estimate of the within group variation and reducing that variance to 0 may artificially increase statistical power.
- In several figures the authors continue using Dunnett's and ANOVA after normalizing the control group to 1, which is not appropriate because the assumptions of normality and equal variance are both broken.
- The cell death quantification in Figure 9 is problematic. This is not so much "cell death" as it is "abnormal nuclear morphology," and examples need to be shown. Was this done manually in a blinded fashion or using an algorithm? We recommend that this data either be substantially reframed or excluded from the manuscript.
- Additional information on the analysis of neurodegeneration in Figure 9 needs to be provided. The methods say that 6-8 fields were quantified for axonal degeneration, but many more dots are plotted in figure 9e and f. Do dots represent individual fields, wells, or batches? Are effects robust across wells and/or batch?

Minor Revisions:

- More information on the no repeat control lines is needed. Have these lines been used before? If not, how did you generate and validate them?
- Figure 1f, $p<0.001$ is written with a comma instead of a period

Reviewer #1

1. *The authors have addressed the concerns raised by the reviewers satisfactorily. The revised manuscript is significantly improved. I recommend this manuscript for the publication without any reservation.*

We thank the reviewer for his/her support.

Reviewer #2

1. *The authors have done a great job of addressing most concerns.*

We thank the reviewer for his/her earlier constructive comments.

2. *There are still major concerns about the statistical approach and we recommend that the authors consult with a statistician. Double normalization as performed by the authors in multiple key figures results in 0 variance in the control group, breaking the assumption of normality required for a Welch's t-test. We might perform a paired t-test based on the authors' description of potentially different experimental conditions, but we strongly recommend that the authors consult a statistician to determine which statistical approaches are appropriate. This is particularly a concern when a small sample size ($n=3$ per group frequently) presumably yields a poor estimate of the within group variation and reducing that variance to 0 may artificially increase statistical power. In several figures the authors continue using Dunnett's and ANOVA after normalizing the control group to 1, which is not appropriate because the assumptions of normality and equal variance are both broken.*

We think Reviewer 2's concerns are reasonable. We consulted with Dr. Oliver King, a colleague and expert on bioinformatics in our school who has previously collaborated with colleagues in the ALS field. Thus, he is familiar with both statistics and ALS disease research. We had several extensive discussions with Dr. King and among ourselves to find the best way to present and perform statistical analysis on our data. We no longer use statistical tests that assume equal between-group variance after normalizing to controls. In the figures we no longer normalize to controls, as Reviewer 2 initially suggested, variation between controls is shown. This makes the assumption of equal variance tenable for the one way ANOVA analyses in Fig 5b,c, Fig 7e, and Fig 8d,e, same as in most other figures. In Fig. 6, where the same neuronal culture was treated with two different small hairpin RNAs, thus they are paired experiments and we used a paired t-test here. In Fig 7c, d, in which the knockout lines appear to have lower variance than the controls, we performed a Welch's t-test, which does not assume equal variance.

3. *The cell death quantification in Figure 9 is problematic. This is not so much "cell death" as it is "abnormal nuclear morphology," and examples need to be shown. Was this done manually in a blinded fashion or using an algorithm? We recommend that this data either be substantially reframed or excluded from the manuscript.*

After careful consideration, we felt that this result does not add much to our story thus decided to exclude this result from the manuscript as the reviewer suggested. We will perform more thorough analysis of the cell death phenotype in the future and include it in another manuscript.

4. *Additional information on the analysis of neurodegeneration in Figure 9 needs to be provided. The methods say that 6-8 fields were quantified for axonal degeneration, but many more dots are plotted in figure 9e and f. Do dots represent individual fields, wells, or batches? Are effects robust across wells and/or batch?*

Each dot (data point) represents individual fields and we analyzed 6-8 fields per differentiation, for a total of 3 different differentiations. This piece of information was already provided in the figure legend. We have now updated the respective Methods section to more clearly describe this experiment.

5. *More information on the no repeat control lines is needed. Have these lines been used before? If not, how did you generate and validate them?*

The no repeat control lines were generated and characterized previously and all the information regarding these lines can be found in Lopez-Gonzalez et al., PNAS 2019 (Ref. #36). We have included this information in the section of “*AFF2* knockout reduces TDP-43 pathology and axonal degeneration in human neurons” in the Results section.

6. *Figure 1f, $p < 0.001$ is written with a comma instead of a period.*

This error has been corrected. Thanks.

REVIEWERS' COMMENTS:

Reviewer #2 (Remarks to the Author):

The authors have sufficiently addressed all the concerns, and I recommend the manuscript for publication.